# Phenothiazine- and Carbazole-Cyanochalcones as Dual Inhibitors of Tubulin Polymerization and Human Farnesyltransferase

**DOI:** 10.3390/ph16060888

**Published:** 2023-06-16

**Authors:** Andreea Zubaș, Alina Ghinet, Amaury Farce, Joëlle Dubois, Elena Bîcu

**Affiliations:** 1Faculty of Chemistry, ‘Alexandru Ioan Cuza’ University of Iasi, Bulevardul Carol I, nr. 11, 700506 Iasi, Romania; andreea.zubas@gmail.com; 2Junia, Health and Environment, Laboratory of Sustainable Chemistry and Health, 59000 Lille, France; 3Institut National de la Santé et de la Recherche Médicale, CHU Lille, Institut Pasteur Lille, U1167–RID-AGE–Facteurs de Risque et Déterminants Moléculaires des Maladies Liées au Vieillissement, University of Lille, 59000 Lille, France; 4Institut National de la Santé et de la Recherche Médicale, CHU Lille, U1286–Infinite-Institute for Translational Research in Inflammation, University of Lille, 59000 Lille, France; amaury.farce@univ-lille.fr; 5Institut de Chimie des Substances Naturelles, UPR2301, CNRS, Centre de Recherche de Gif, 91190 Gif-sur-Yvette, France; duboisdacquigny@wanadoo.fr

**Keywords:** tubulin, inhibitor, dual, farnesyltransferase, cancer, cyanochalcone, sonication

## Abstract

In the search for innovative approaches to cancer chemotherapy, a chemical library of 49 cyanochalcones, **1a-r**, **2a-o**, and **3a-p**, was designed as dual inhibitors of human farnesyltransferase (FTIs) and tubulin polymerization (MTIs) (FTIs/MTIs), two important biological targets in oncology. This approach is innovative since the same molecule would be able to interfere with two different mitotic events of the cancer cells and prevent these cells from developing an emergency route and becoming resistant to anticancer agents. Compounds were synthesized by the Claisen–Schmidt condensation of aldehydes with *N*-3-oxo-propanenitriles under classical magnetic stirring and under sonication. Newly synthesized compounds were screened for their potential to inhibit human farnesyltransferase, tubulin polymerization, and cancer cell growth in vitro. This study allowed for the identification of 22 FTIs and 8 dual FTIs/MTIs inhibitors. The most effective molecule was carbazole-cyanochalcone **3a**, bearing a 4-dimethylaminophenyl group (IC_50_ (h-FTase) = 0.12 µM; IC_50_ (tubulin) = 0.24 µM) with better antitubulin activity than the known inhibitors that were previously reported, phenstatin and (-)-desoxypodophyllotoxin. The docking of the dual inhibitors was realized in both the active site of FTase and in the colchicine binding site of tubulin. Such compounds with a dual inhibitory profile are excellent clinical candidates for the treatment of human cancers and offer new research perspectives in the search for new anti-cancer drugs.

## 1. Introduction

The majority of new cancer cases detected worldwide each year are treated with anti-cancer drugs. Unfortunately, many cancer mutations become resistant to these drugs. An alternative consists of using cocktails of drugs acting on various biological targets of interest in oncology to overcome resistant cells. This type of approach can give good results, but it often leads to a large increase in side effects. Advances in the field of cell biology have allowed for the identification of new targets for the treatment of cancer, opening up new therapeutic perspectives. Another alternative to avoid the use of multiple anticancer drugs to stop cancer cell growth proliferation is to use a single compound acting on two different biological targets [1,2]. Our approach concerns the inhibition of two aspects of cell division occurring at two very different times in the life of a cancer cell: one involving farnesyltransferase and the other involving tubulin. Designed compounds target farnesyltransferase, a zinc metalloenzyme, and also inhibit tubulin polymerization. Tubulin is involved in cell proliferation due to its ability to polymerize and form microtubules, key components of the cytoskeleton. This protein is the target of a large panel of small molecules that interfere with the dynamics of its polymerization or depolymerization. Most of them bind to the laulimalide, maytansine, taxane/epothilone, vinca alkaloid, and colchicine sites [3]. By interacting with tubulin and microtubules, the tubulin polymerization inhibitors block cells in mitosis; this results in their accumulation in the G2/M phase of the cancer cell cycle. For the design of our potential dual inhibitors, two known strong inhibitors of tubulin polymerization, combretastatin A-4 (CA-4) (**I**, Figure 1) and phenstatin (**II**, Figure 1), were considered as reference molecules. Most of the modifications previously described in the structure of CA-4 and phenstatin involved either the ethylenic or carbonyl bridge or the methoxyphenol B ring. However, the 3,4,5-trimethoxyphenyl group (ring A) has long been kept intact, as it was considered essential for cytotoxic activity as well as the inhibition of tubulin polymerization. Our group previously described that a completely different A ring consisting of a phenothiazine unit can successfully replace the 3,4,5-trimethoxyphenyl of phenstatin and provide effective tubulin polymerization (e.g., compounds **III** and **IV**, Figure 1) [4,5].

The other target of molecules from this study was human protein farnesyltransferase (FTase). FTase is a heterodimeric metalloenzyme that belongs to the protein prenyl transferase family and is composed of two subunits: α (48 kDa) and β (45 kDa). Farnesylation is a post-translational modification occurring in several cell signaling proteins such as small GTPases, including the oncogenic Ras proteins that play a fundamental role in cancer cell growth and division [6]. FTase catalyzes the transfer of a farnesyl group (C_15_) from farnesyl pyrophosphate or farnesyl diphosphate (FPP) to the free thiol group of a cysteine residue embedded in the C-terminal CaaX motif of proteins where C is a cysteine, a is an aliphatic amino acid, and X is a serine, a methionine, an alanine, or a glutamine [7]. Preventing the farnesylation process may constitute an approach in the treatment of cancers, and, therefore, farnesyltransferase inhibitors (FTIs) were developed for anticancer therapy, and diverse compounds with druglike properties are available [7,8,9,10,11]. The use of farnesyltransferase inhibitors was disappointing in clinical trials for cancer treatment. Indeed, even if FTase is completely inhibited, a bypass is always possible for the cancerous cell. This alternative path involves a protein very similar to FTase, which is geranylgeranyltransferase I (GGTase-I) [12]. However, proving the effectiveness of dual compounds FTIs/MTIs, which are inhibitors of FTase (FTIs) and of tubulin polymerization (MTIs), may lead to an innovative approach for the design of new anti-cancer compounds.

Several associations between FTIs and MTIs were described in the literature. The association of lonafarnib (**SCH66336**, compound **V**, Figure 1) with paclitaxel resulted in an enhanced cytotoxic effect in ovarian cancer cells in vitro and in vivo [13]. The same association of lonafarnib/paclitaxel (Taxol) or lonafarnib/docetaxel (Taxotere) is synergistic in vivo in NCI-460 lung cancer cells, and lonafarnib could also be used by patients who develop resistance to taxanes. Another FTI (**FTI-277**, compound **VI**, Figure 1) displayed synergistic effect with paclitaxel or docetaxel in cells resistant to paclitaxel [14].
Figure 1Previously described potent inhibitors of tubulin polymerization (CA-4 (**I**), phenstatin (**II**), phenothiazines (**III**,**IV**)) [4,5] and of human farnesyltransferase (Lonafarnib (SCH66336) (**V**) [13] and FTI-277 (**VI**) [14]) investigated as anticancer compounds.
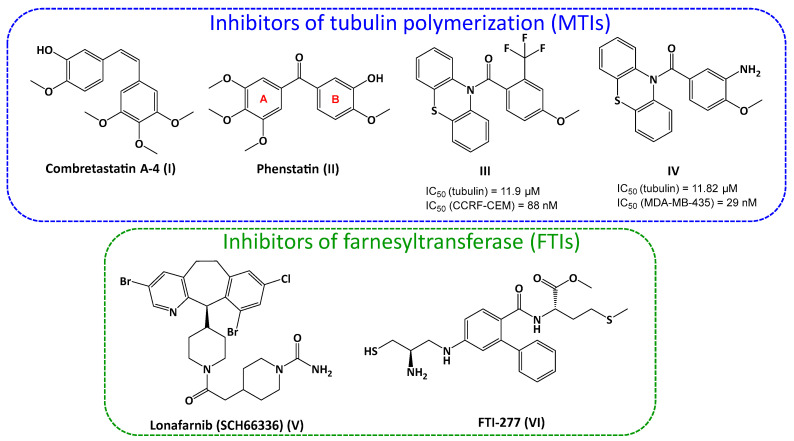


Based on all these previous findings presented above, a new series of anticancer agents, dual inhibitors of farnesyltransferase and tubulin polymerization FTIs/MTIs, were developed in this study (compounds **1a-r**, **2a-o**, and **3a-p**, Figure 2). These compounds are not prodrugs or a combination of two known specific inhibitors, rather they are original structures rationally designed from previous studies and the literature data. These compounds, **1a-r**, **2a-o**, and **3a-p** (Figure 2), share a common bridge between the A and B rings, which is a cyanochalcone group. Another particularity of these target molecules is phenothiazine (compounds **1a-r**), 2-methylthiophenothiazine (compounds **2a-o**), and carbazole (compounds **3a-p**) as the A ring. The literature analysis allowed for the identification of some similar phenothiazine-cyanochalcones that are nitric oxide (NO) inhibitors, preventing diseases mediated by lipid peroxidation (compound **VII**, Figure 2) [15], or display antibacterial activity against the Gram-positive bacteria *Bacillus subtilis* (compounds **VIII** and **IX**, Figure 2) or the Gram-negative bacteria *Escherichia coli* (compound **VIII**, Figure 2) [16]. To the best of our knowledge, there is no phenothiazine- or carbazole-cyanochalcone described for its anticancer properties to date. Only one similar cyanoacetamide that integrated phenothiazine (compound **XII**, Figure 2) displayed a modest antitumor activity in vitro, inhibiting the growth of SW1990 (86.7%) and AsPC1 (74.68%) pancreatic tumor cells at 100 µM concentration [17]. On the contrary, chalcones with a phenothiazine (compounds **X** and **XI**, Figure 2) or carbazole (compound **XIII**, Figure 2) or azacarbazole ring (compounds **XIV** and **XV**, Figure 2) were previously reported for their antitumor activity. Phenothiazine-chalcone **X** inhibited human FTase with an excellent IC_50_ value of 9 nM [10], while phenothiazine derivative **XI** was active against Hep-G2 cells [18]. Carbazole chalcone **XIII** was more effective in inhibiting HL-60 leukemia cells with a submicromolar range (IC_50_ (HL-60) = 0.22 µM, Figure 2). Azacarbazole-chalcone **XIV** inhibited the growth of MCF-7 breast cancer cell lines in the low micromolar range [19], while azacarbazole-chalcone **XV** displayed similar IC_50_ value in melanoma B-16 cells (Figure 2) [20]. However, increasing interest was shown in the recent literature, reporting the development of phenothiazine hybrids with potential medicinal interest and other properties [21,22,23].

## 2. Results and Discussion

### 2.1. Synthetic Strategy

The cyanochalcones (**1a-r**, **2a-o**, and **3a-p**) of this study were prepared by the Claisen–Schmidt condensation of the corresponding *N*-3-oxo-propanenitriles **4a**, **4b**, and **5** and (hetero)aryl aldehydes **8**–**36** (Figure 1). *N*-3-oxo-propanenitriles **4a**, **4b**, and **5** were obtained by treating phenothiazine **6a**, 2-methylthiophenothiazine **6b**, and carbazole **7** with the mixed anhydride of acetic acid and cyanoacetic acid obtained in situ from an equimolar mixture of cyanoacetic acid and acetic anhydride (Figure 1). The resulting *N*-3-oxo-propanenitriles **4a** and **5** were previously described, and their physicochemical characterization corresponded to that reported in the literature [24,25]. Next, the key condensation reaction was conducted under classical magnetic stirring, and, based on the effective results obtained previously for the Claisen–Schmidt condensation of (hetero)aryl ketones with (hetero)aryl aldehydes [11], also under sonication of the mixture instead of classical magnetic stirring. In the classical magnetic stirring procedure (procedure A, Figure 1), piperidine and glacial acetic acid were used as catalysts and ethanol or acetonitrile as solvents. The reaction media were stirred under reflux for the phenothiazine derivatives and at rt for the carbazole derivatives. The sonication method (procedure B, Figure 1) used LiOH as a base and ethanol as a solvent. Both procedures, applied to the corresponding *N*-3-oxo-propanenitriles **4a**, **4b**, and **5** and aldehydes **8**–**36**, allowed for the obtainment of a large panel of 49 cyanochalcones in medium to high yields (40–87%) (see Table 1 and [Fig pharmaceuticals-16-00888-ch001] and [Fig pharmaceuticals-16-00888-ch002]). All compounds were obtained as *E*-isomers. No trace of *Z*-isomers was detected in this series. In order to conduct the greenest and least energy-consuming synthetic method possible for obtaining cyanochalcones, the synthesis of the same compound was carried out following the two procedures. This resulted in comparable yields, but the reaction time was significantly reduced from hours to minutes by sonication. As an example, the phenothiazine derivative **1m** was obtained in 74% yield under sonication and in 67% yield under magnetic stirring, but after only 2 min (procedure B) against 24 h (procedure A) ([Fig pharmaceuticals-16-00888-ch001]). Consequently, a major part of 2-methylthiophenothiazine-cyanochalcones **2a**, **2c-f**, **2h-j**, and **2o** was further synthesized under sonication in less than 90 s (Table 1). Under the standard conditions of procedure A, the synthesis of these latter compounds would have required refluxing ethanol or acetonitrile for 24 h. Carbazole-cyanochacones **3a-p** were synthesized using procedure A, since they could be easily obtained at room temperature (Figure 1 and [Fig pharmaceuticals-16-00888-ch002]).

### 2.2. Biological Evaluation

Synthesized compounds were further evaluated in vitro on the two biological targets: farnesyltransferase and tubulin. The results of these biological evaluations are presented in Table 2. Potential inhibitors were first screened at a high concentration (100 µM), and only compounds that generally inhibited more than 60% of the proteins were selected for IC_50_ calculation. Dimethylsulfoxide (DMSO) was used as a negative reference, while phenstatin (**II**) and (-)-desoxypodophyllotoxin were positive references for the tubulin polymerization assay, and FTI-276 was the positive reference for the evaluation on human FTase (Table 2). FTI-277 (**VI**, Figure 1) is a prodrug of FTI-276, the latter being more affine to FTase than parent FTI-277. Interestingly, a large part of the tested cyano-chalcones inhibited FTase and presented a moderate to potent effect (IC_50_ values ranging from tens of micromoles (e.g., phenothiazine **1q**: IC_50_ (h-FTase) = 44.85 µM) on submicromolar (e.g., carbazole **3a**: IC_50_ (h-FTase) = 0.12 µM) concentrations (Table 2). In the tubulin polymerization assay, fewer compounds were found to be tubulin polymerization compounds, but two of these inhibitors significantly outperformed the potencies of the positive references phenstatin (**II**) and (-)-desoxypodophyllotoxin (e.g., compare phenothiazine **1l**: IC_50_ (tubulin) = 0.71 µM or carbazole **3a**: IC_50_ (tubulin) = 0.24 µM with the positive reference phenstatin (**II**), IC_50_ (tubulin) = 3.43 µM, and with (-)-desoxypodophyllotoxin: IC_50_ (tubulin) = 1.76 µM, Table 2). Moreover, four carbazole-cyanochalcones **3b**, **3i**, **3j**, and **3l** displayed similar inhibitory activity to that of the reference inhibitors (Table 2). Considering both biological evaluations, it can be concluded that several compounds inhibited the two biological targets of interest and may be considered as dual FTIs/MTIs. This is the case for one phenothiazine-cyanochalcone, **1l**, and seven carbazole-cyanochalcones, **3a**, **3b**, **3d**, **3e**, **3i**, **3j**, and **3l**. The carbazole-cyanochalcone **3a** was the most potent inhibitor discovered in this study, inhibiting both targets and presenting submicromolar IC_50_ (0.12 µM for h-FTase and 0.24 µM for tubulin polymerization, respectively). The corresponding phenothiazine **1c** was not active (Table 2). Now, looking at the chemical structures of phenothiazine **1l** and carbazole **3j**, they have the same classical B-ring as CA-4 (**I**, Figure 1) and phenstatin (**II**, Figure 1). This ring seems important to the biological activity against tubulin, especially in the phenothiazine series. Its replacement by other substituents in the phenothiazine cyanochalcones (**1a-k**, **1m-r**) abolished the inhibitory effect (Table 2). On the contrary, in the series of carbazoles, the replacement of the classical 3′-hydroxy-4′-methoxyphenyl B ring was tolerated. The 4-dimethylaminophenyl group in compound **3a** had the best modulation in the current study. Moreover, the reverse substitution of the classical B ring (3′-methoxy-4′-hydroxyphenyl in compound **3l** instead of 3′-hydroxy-4′-methoxyphenyl in compound **3j**) conserved antitubulin activity. The 3′-fluoro-4′-methoxyphenyl substitution in compound **3i** was also tolerated, while the substitution of the 3′-fluoro by a 3′-chloro in compound **3h** resulted in the loss of the biological activity (Table 2). The suppression of the 4′-methoxy group in carbazole **3d** dramatically decreased the antitubulin potential (compare carbazole **3d** (IC_50_ (tubulin) = 69.8 µM) with **3j** (IC_50_ (tubulin) = 2.92 µM), Table 2). The 4′-nitro substitution in carbazole **3e** conserved an inhibitory potential (IC_50_ (tubulin) = 10.45 µM), Table 2) but was significantly reduced compared to that in carbazole **3a**.

To better visualize the distribution of the cyanochalcones from this study into selective human FTase inhibitors or dual inhibitors, their clustering was realized using POWER BI Desktop software version 2.117.984.0.(Figure 3). Eight dual FTIs/MTIs (cluster in pink, Figure 3) and twenty-two inhibitors of human farnesyltransferase (cluster in green, Figure 3) were found. The dual inhibitors were generally carbazole cyanochalcones (**3a**, **3b**, **3d**, **3e**, **3i**, **3j**, and **3l**), except for phenothiazine **1l**, which also displayed dual inhibitory potential. The phenothiazine cyanochalcones displayed an FTIs profile.

2-Methylthiophenothiazine-cyanochalcones **2a-o** were submitted to the National Cancer Institute (NCI) and were further selected for biological evaluation based on their panel of 60 cancer cell lines in vitro. Thus, 2-methylthiophenothiazines **2a-o** were tested at a single dose of 10 µM in the full NCI-60 panel. The one-dose data are reported as an average of the growth inhibition percentage of the treated cells (Table 3 and Appendix A). Only three compounds, **2k**, **2l**, and **2o**, from the series were active. Cyanochalcone **2k**, bearing an indole unit, was the most active among the tested compounds, inhibited the growth of the majority of the tested cancer cells by more than 50% (Table 3), and even reached an inhibition greater than 90% in HL-60(TB) leukemia, NCI-H522 non-small cell lung cancer, and SF-539 and SNB-75 CNS cancer cell lines. The two other 2-methylthiophenothiazines, **2l** and **2o**, were significantly less active. Compound **2l**, bearing a 5-methoxyindole unit, displayed moderate activity in CCRF-CEM leukemia (49% inhibition), KM12 colon cancer (49% inhibition), PC-3 prostate cancer (60% inhibition), T-47D breast cancer (57% inhibition), and CAKI-1 renal cancer (49% inhibition) cells (Table 3). Finally, compound **2o**, obtained from trans-cinnamaldehyde, was the least active and only inhibited the growth of MCF7 (59% inhibition) and MDA-MB-468 (60% inhibition) breast cancer cells (Table 3).

### 2.3. Molecular Docking of Dual FTIs/MTIs Inhibitors

The docking study was next realized on the dual FTIs/MTIs inhibitors identified in this study in the active site of FTase and in the colchicine binding site of tubulin to understand their binding mode. All the data obtained for the docking of the eight dual inhibitors are available in the Appendix A). The structure of the human FTase was obtained from its complexed X-ray crystal structure in the RCSB Protein Data Bank (1LD7) with FPP and the inhibitor molecule described by Bell et al. [26]. The flexible docking of FTIs into the enzyme active site was performed using GOLD 5.1 [27]. The binding site was defined by a 10 Å sphere around the cocrystallized ligand of 1LD7, and 30 poses were generated for each compound using GoldScore as the scoring function. The solutions were selected by checking the superimposition of the poses, keeping the most representative of the largest clusters. The protocol used for the docking of the selected molecules in the tubulin binding site (colchicine site) was realized as previously reported [4].

#### 2.3.1. Docking on FTase

All the investigated compounds fit well in the pocket (Figure 4 and Appendix A). The largest number, consisting of compounds **1l**, **3b**, **3e**, **3i**, **3j**, and **3l**, has their tricyclic group toward the entry of the pocket, and all form interactions with the zinc ion. Tyr 601 is involved in a hydrogen bond with **1l**, **3j**, and **3l** (Figure 4a,d and Appendix A).

The tricyclic part of compounds **3a** (Figure 4b) and **3d** (Figure 4c) is superimposed, though more toward Trp 407 than the other molecules. Moreover, only **3d** can interact with zinc, as the fluorobenzene moiety of **3a** is also oriented toward Trp 407, stabilizing the compound by a distant and not optimal double stacking with it.

#### 2.3.2. Docking in the Tubulin Binding Site (Colchicine Site)

The reference, phenstatin **II**, binds to the backbone of Ala 732, while mostly being in a hydrophobic region (Figure 5f). Compounds **3b** and **3a** (Figure 5c) and the 40% highest score of compound **1l** (Figure 5a) all form a cluster closer to the entry of the binding site, where they can form a hydrogen bond with Asn 682. Compounds **3d**, **3e** (Figure 5d), and **3i** all superimpose well in a conformation deeper in the pocket, with the cyano able to form a hydrogen bond with the Ser 168 of α tubuline.

A third cluster is formed by compounds **3j** and **3l** (Figure 5e) and the 60% lowest score of compound **1l** (Figure 5b), with a better occupation of the deepest part of the pocket than phenstatin but fully lacking any hydrogen bond and counting on their hydrophobic fitting to stay in the binding site.

### 2.4. Conclusions

In this study, a chemical collection of 49 cyanochalcones, **1a-r**, **2a-o**, and **3a-p**, decorated with phenothiazine, 2-methylthiophenothiazine, and carbazole rings was designed and synthesized by the Claisen–Schmidt condensation of the corresponding *N*-3-oxo-propanenitriles and aldehydes. The synthetic procedure was realized either under classical magnetic stirring and heating or under sonication of the medium. The ultrasound-assisted condensation allowed for a reduction in the reaction time from hours to minutes, especially for the synthesis of phenothiazine cyanochalcones. The therapeutic strategy described in this report was used to obtain dual FTIs/MTIs inhibitors. This approach is innovative, since the same molecule would be able to interfere with two different mitotic events of cancer cells and prevent these cells from developing an emergency route and becoming resistant to anticancer agents. Synthesized compounds were evaluated in vitro on human farnesyltransferase, on tubulin polymerization, and on the NCI-60 cancer cell lines panel. Phenothiazine derivatives proved to be inhibitors of human FTase, while carbazole derivatives displayed dual inhibition of FTase and tubulin polymerization. Of interest, phenothiazine cyanochalcone **1l** and carbazole cyanochalcone **3a** displayed better antitubulin activity than that of the known inhibitors previously reported: phenstatin **II** and (-)-desoxypodophyllotoxin. Carbazole derivatives were more active than the phenothiazine analogues. This study allowed for the identification of 22 FTIs and 8 dual FTIs/MTIs inhibitors. The most effective molecule was carbazole-cyanochalcone **3a** bearing a 4-dimethylaminophenyl group (IC_50_ (h-FTase) = 0.12 µM; IC_50_ (tubulin) = 0.24 µM). The docking of the dual inhibitors was realized both in the active site of FTase and in the colchicine binding site of tubulin and allowed for the visualization of their binding modes. The biological evaluation of these promising dual inhibitors in several cancer cell lines and the evaluation of their pharmacokinetic parameters will be realized in due course. Such compounds with a dual inhibitory profile are excellent clinical candidates for the treatment of human cancers and offer new research perspectives in the search for new anti-cancer drugs.

## 3. Materials and Methods for Synthesis and Characterizations

Starting materials were commercially available and were used without further purification (suppliers: Carlo Erba Reagents S.A.S., Val de Reuil, France, Thermo Fisher Scientific Inc., Illkirch-Graffenstaden, France, and Sigma-Aldrich Co., Saint-Quentin-Fallavier, France). Ultrasound-mediated reactions were realized using Q700S apparatus (QSonica, LLC, Newton, MA, USA) and CL-334 model probe. Melting points were measured on a MPA 100 OptiMelt^®^ apparatus (Stanford Research Systems, Sunnyvale, CA, USA) and a KRÜSS Optronic KSP1N apparatus (A.KRÜSS Optronic GmbH, Hamburg, Germany) and were uncorrected. Nuclear magnetic resonance (NMR) spectra were acquired at 500 MHz for ^1^H NMR and at 125 MHz for ^13^C NMR on a Bruker Avance III spectrometer (Bruker, Mannheim, Germany) and at 400 MHz for ^1^H NMR and at 100 MHz for ^13^C NMR on a Varian 400-MR spectrometer (Varian, Les Ulis, France) with tetramethylsilane (TMS) as internal standard, at room temperature (RT). All spectra were realized using deuterated solvents (CDCl_3_ 99.8%D + 0.03% TMS V/V or DMSO-d_6_ 99.8%D + 0.03% TMS V/V), purchased from Eurisotop, Saint-Aubin, France. The calibration was realized using TMS pic as the 0.00 ppm value in the registered spectra. Chemical shifts (δ) were expressed in ppm relative to TMS. Splitting patterns were designed: s, singlet; d, doublet; dd, doublet of doublets; t, triplet; td, triplet of doublets; q, quadruplet; quint, quintuplet; m, multiplet; sym m, symmetric multiplet; br s, broaden singlet; br t, broaden triplet. Coupling constants (*J*) were reported in hertz (Hz). Thin-layer chromatography (TLC) was realized on Macherey Nagel silica gel plates with fluorescent indicator and were visualized under a UV lamp at 254 nm and 365 nm. Column chromatography was performed with a CombiFlash Rf Companion (Teledyne-Isco System, Serlabo Technologies, Entraigues sur la sorgues, France) using RediSep packed columns. IR spectra were recorded on a FT-IR Bruker Tensor 27 Spectrometer (Bruker, MA, USA) or a Cary 630 FT-IR Spectrometer (Agilent Technologies, Les Ulis, France). Elemental analyses (C, H, N, S) of new compounds were determined on a Thermo Electron apparatus by “Pôle Chimie Moléculaire-Welience”, Faculté de Sciences Mirande, Université de Bourgogne, Dijon, France.

### 3.1. General Procedure for the Synthesis of N-Cyanoacetyl-phenothiazines ***4a*** and ***4b*** and N-Cyanoacetyl-carbazole 5

A mixture of cyanoacetic acid (2 equiv.) and acetic anhydride (2 equiv.) was stirred at 50–80 °C. After complete solubilization, phenothiazine derivative **6a** or **6b** or carbazole **7** (1 equiv.) was added, and the mixture was stirred at 100 °C for 1 h. The precipitate formed was filtered and purified by recrystallization from ethanol to provide pure product **4a**, **4b**, or **5**.

The physicochemical characterization of compounds **4a** and **5** corresponded to that previously described in the literature [24,25].

#### 3-(2-(Methylthio)-10H-phenothiazin-10-yl)-3-oxopropanenitrile (**4b**)

The general procedure was used with cyanoacetic acid (6.93 g, 81.6 mmol), acetic anhydride (8.34 g, 81.6 mmol), and 2-(methylthio)-10*H*-phenothiazine (10.00 g, 40.8 mmol) to obtain pure compound **4b** (10.58 g, 33.8 mmol, 83% yield) as a mint-green solid; mp 127–128 °C (EtOH); ^1^H NMR (CDCl_3_, 500 MHz) δ ppm: 2.51 (s, 3H, SC*H*_3_), 3.60 (s, 2H, C*H*_2_), 7.17 (dd, *J* = 7.5, 2.0 Hz, 1H, Ar*H*), 7.30 (t, *J* = 7.5 Hz, 1H, Ar*H*), 7.38 (dt, *J* = 7.5, 1.5 Hz, 2H, Ar*H*), 7.42 (br s, 1H, Ar*H*), 7.46–7.56 (m, 2H, Ar*H*); ^13^C NMR (CDCl_3_, 125 MHz) δ ppm: 61.0 (SCH_3_), 26.3 (CH_2_), 113.7 (CN), 123–133 (7CH+C), 137.4 (2C), 138.2 (2C), 161.1 (C=O); IR ν (cm^−1^): 2218, 1668, 1446, 1363, 1316, 1257, 1126, 990, 762, 730. Anal. Calcd for C_16_H_12_N_2_OS_2_: C, 61.51; H, 3.87; N, 8.97. Found: C, 61.37; H, 3.64; N, 8.81%.

### 3.2. General Procedure for the Synthesis of Chalcone Analogues (***1a-e***, ***1g-r***, ***2b***, ***2g***, ***2k-n***, and ***3a-p***) by Claisen–Schmidt Condensation under Classical Magnetic Stirring)—Procedure A

To a solution of phenothiazine derivative **4a** or **4b** or carbazole derivative **5** (1.0 equiv.) and an aromatic/heteroaromatic aldehyde (1.2 equiv.) in ethanol or acetonitrile, piperidine (3–4 drops) and glacial acetic acid (1–3 drops) were added dropwise, and the resulting solution was stirred at reflux for 3–24 h. The reaction was monitored by TLC (EtOAc:Cyclohexane) until complete consumption of starting substrate **4a**, **4b**, or **5**. The formed precipitate was filtered, washed with ethanol, and purified by recrystallization from ethanol or by flash column chromatography (silica gel 60 (0.063–0.200 mm, 60 Å), mobile phase: gradient cyclohexane/EtOAc 100/0 to 0/100) to obtain pure cyanochalcone (**1a-e**, **1g-r**, **2b**, **2g**, **2k-n**, and **3a-p**).

#### 3.2.1. (E)-2-(10H-Phenothiazine-10-carbonyl)-3-phenylacrylonitrile (**1a**)

General procedure A was used with 3-oxo-3-(10*H*-phenothiazin-10-yl)propanenitrile **4a** (0.67 g, 2.51 mmol), benzaldehyde **8** (0.32 g, 3.02 mmol), piperidine (3 drops), and glacial acetic acid (3 drops) in 10 mL acetonitrile to obtain pure **1a** (0.68 g, 1.92 mmol, 76% yield) as a yellow solid; mp > 250 °C (EtOH); ^1^H NMR (CDCl_3_, 400 MHz) δ ppm: 7.23–7.36 (m, 4H, Ar*H*), 7.39–7.52 (m, 5H, Ar*H*), 7.62 (d, *J* = 7.6 Hz, 2H, Ar*H*), 7.81 (d, *J* = 7.6 Hz, 2H, Ar*H*), 8.01 (s, 1H, =C*H*); ^13^C NMR (CDCl_3_, 100 MHz) δ ppm: 107.3 (C), 114.1 (CN), 126.3 (2CH), 127.2 (2CH), 127.4 (2CH), 128.1 (2CH), 129.1 (2CH), 130.4 (2CH), 132.0 (C), 132.5 (CH), 132.8 (2C), 138.2 (2C), 153.8 (=CH), 162.0 (C=O). IR ν (cm^−1^): 2207, 1661, 1590, 1327, 1182, 807, 759. Anal. Calcd for C_22_H_14_N_2_OS: C, 74.55; H, 3.98; N, 7.90. Found: C, 74.69; H, 4.02; N, 8.11%.

#### 3.2.2. (E)-2-(10H-Phenothiazine-10-carbonyl)-3-(p-tolyl)acrylonitrile (**1b**)

General procedure A was used with 3-oxo-3-(10*H*-phenothiazin-10-yl)propanenitrile **4a** (0.50 g, 1.87 mmol), 4-methylbenzaldehyde **9** (0.26 g, 2.16 mmol), piperidine (3 drops), and glacial acetic acid (3 drops) in 15 mL acetonitrile to obtain pure **1b** (0.45 g, 1.22 mmol, 65% yield) as a yellowish solid; mp 221–223 °C (EtOH); ^1^H NMR (CDCl_3_, 500 MHz) δ ppm: 2.40 (s, 3H, C*H*_3_), 7.22–7.35 (m, 6H, Ar*H*), 7.49 (dd, *J* = 8.0, 1.0 Hz, 2H, Ar*H*), 7.61 (d, *J* = 8.0 Hz, 2H, Ar*H*), 7.73 (d, *J* = 8.0 Hz, 2H, Ar*H*), 8.00 (s, 1H, =C*H*); ^13^C NMR (CDCl_3_, 125 MHz) δ ppm: 21.9 (CH_3_), 106.0 (C), 114.6 (CN), 126.4 (2CH), 127.4 (2CH), 127.5 (2CH), 128.2 (2CH), 129.6 (C), 130.0 (2CH), 130.7 (2CH), 132.9 (2C), 138.5 (2C), 143.9 (C), 154.1 (=CH), 162.4 (C=O); IR ν (cm^−1^): 2207, 1660, 1588, 1460, 1326, 1262, 1181, 1125, 1032, 951, 807, 766, 665, 603. Anal. Calcd for C_23_H_16_N_2_OS: C, 74.98; H, 4.38; N, 7.60. Found: C, 75.31; H, 4.65; N, 7.91%.

#### 3.2.3. (E)-3-(4-(Dimethylamino)phenyl)-2-(10H-phenothiazine-10-carbonyl)acrylonitrile (**1c**)

General procedure A was used with 3-oxo-3-(10*H*-phenothiazin-10-yl)propanenitrile **4a** (0.25 g, 0.94 mmol), 4-(dimethylamino)benzaldehyde **10** (0.17 g, 1.13 mmol), piperidine (3 drops), and glacial acetic acid (1 drop) in 10 mL ethanol to obtain pure **1c** (0.31 g, 0.78 mmol, 85% yield) as a yellow solid; mp > 250 °C (EtOH); ^1^H NMR (CDCl_3_, 500 MHz) δ ppm: 3.07 (s, 6H, 2C*H*_3_), 6.65 (d, *J* = 9.0 Hz, 2H, Ar*H*), 7.22–7.28 (m, 2H, Ar*H*), 7.30 (td, *J* = 7.5, 1.5 Hz, 2H, Ar*H*), 7.48 (dd, *J* = 7.5, 1.5 Hz, 2H, Ar*H*), 7.65 (dd, *J* = 7.5, 1.5 Hz, 2H, Ar*H*), 7.81 (d, *J* = 9.0 Hz, 2H, Ar*H*), 7.97 (s, 1H, =C*H*); ^13^C NMR (CDCl_3_, 125 MHz) δ ppm: 40.3 (2CH_3_), 98.6 (C), 111.6 (2CH), 116.2 (CN), 120.2 (C), 126.5 (2CH), 127.1 (2CH), 127.3 (2CH), 128.1 (2CH), 132.9 (2C), 133.5 (2CH), 139.1 (2C), 153.3 (C), 154.6 (=CH), 163.7 (C=O); IR ν (cm^−1^): 2202, 1665, 1613, 1571, 1531, 1460, 1384, 1319, 1263, 1181, 1132, 1063, 1029, 946, 889, 810, 753, 664. Anal. Calcd for C_24_H_19_N_3_OS: C, 72.52; H, 4.82; N, 10.57. Found: C, 72.77; H, 4.93; N, 10.68%.

#### 3.2.4. (E)-3-(4-Methoxyphenyl)-2-(10H-phenothiazine-10-carbonyl)acrylonitrile (**1d**)

General procedure A was used with 3-oxo-3-(10*H*-phenothiazin-10-yl)propanenitrile **4a** (0.50 g, 1.87 mmol), 4-methoxylbenzaldehyde **11** (0.31 g, 2.25 mmol), piperidine (4 drops), and glacial acetic acid (1 drop) in 10 mL ethanol to obtain pure **1d** (0.57 g, 1.48 mmol, 79% yield) as a yellow solid; mp 230–231 °C (EtOH); ^1^H NMR (CDCl_3_, 500 MHz) δ ppm: 3.78 (s, 3H, OC*H*_3_), 6.87 (d, *J* = 9.0 Hz, 2H, Ar*H*), 7.18–7.30 (m, 4H, Ar*H*), 7.42 (dd, *J* = 7.5, 2.0 Hz, 2H, Ar*H*), 7.55 (dd, *J* = 7.5, 2.0 Hz, 2H, Ar*H*), 7.75 (d, *J* = 9.0 Hz, 2H, Ar*H*), 7.88 (s, 1H, =C*H*); ^13^C NMR (CDCl_3_, 125 MHz) δ ppm: 55.0 (OCH_3_), 102.7 (C), 114.0 (2CH), 114.1 (C), 114.2 (CN), 124.2 (C), 125.7 (2CH), 125.8 (C), 126.6 (2CH), 126.7 (2CH), 127.4 (2CH), 131.9 (C), 132.2 (2CH), 137.7 (2C), 153.0 (=CH), 162.6 (C=O); IR ν (cm^−1^): 2213, 2156, 1677, 1592, 1511, 1459, 1427, 1320, 1259, 1180, 1121, 1060, 1019, 926, 891, 828, 757, 726, 663. Anal. Calcd for C_23_H_16_N_2_O_2_S: C, 71.85; H, 4.19; N, 7.29. Found: C, 72.05; H, 4.30; N, 7.61%.

#### 3.2.5. (E)-3-(4-Bromophenyl)-2-(10H-phenothiazine-10-carbonyl)acrylonitrile (**1e**)

General procedure A was used with 3-oxo-3-(10*H*-phenothiazin-10-yl)propanenitrile **4a** (0.25 g, 0.94 mmol), 4-bromobenzaldehyde **12** (0.21 g, 1.13 mmol), piperidine (4 drops), and glacial acetic acid (1 drop) in 10 mL ethanol to obtain pure **1e** (0.29 g, 0.67 mmol, 71% yield) as a yellow solid; mp 233–235 °C (EtOH); ^1^H NMR (CDCl_3_, 500 MHz) δ ppm: 7.28–7.37 (m, 4H, Ar*H*), 7.51 (dd, *J* = 7.5, 1.5 Hz, 2H, Ar*H*), 7.55–7.63 (m, 4H, Ar*H*), 7.68 (d, *J* = 7.5 Hz, 2H, Ar*H*), 7.96 (s, 1H, =C*H*); ^13^C NMR (CDCl_3_, 125 MHz) δ ppm: 108.0 (C), 114.1 (CN), 126.4 (2CH), 127.4 (2CH), 127.5 (C), 127.7 (2CH), 128.3 (2CH), 131.0 (C), 131.8 (2CH), 132.6 (2CH), 132.9 (2C), 138.2 (2C), 152.6 (=CH), 161.8 (C=O); IR ν (cm^−1^): 2218, 1679, 1584, 1489, 1460, 1407, 1323, 1262, 1188, 1076, 1009, 816, 756. Anal. Calcd for C_22_H_13_BrN_2_OS: C, 60.98; H, 3.02; N, 6.46. Found: C, 61.19; H, 3.34; N, 6.62%.

#### 3.2.6. (E)-3-(2,5-Dimethoxyphenyl)-2-(10H-phenothiazine-10-carbonyl)acrylonitrile (**1g**)

General procedure A was used with 3-oxo-3-(10*H*-phenothiazin-10-yl)propanenitrile **4a** (0.30 g, 1.13 mmol), 2,5-dimethoxybenzaldehyde **19** (0.23 g, 1.38 mmol), piperidine (4 drops), and glacial acetic acid (1 drop) in 15 mL ethanol to obtain pure **1g** (0.35 g, 0.84 mmol, 75% yield) as a yellow solid; mp 184–186 °C (EtOH); ^1^H NMR (CDCl_3_, 500 MHz) δ ppm: 3.74 (s, 3H, OC*H*_3_), 3.84 (s, 3H, OC*H*_3_), 6.87 (d, *J* = 9.0 Hz, 1H, Ar*H*), 7.03 (dd, *J* = 9.0, 2.0 Hz, 1H, Ar*H*), 7.27–7.36 (m, 4H, Ar*H*), 7.61–7.67 (m, 3H, Ar*H*), 7.90 (dd, *J* = 7.5, 2.0 Hz, 2H, Ar*H*), 8.50 (s, 1H, =C*H*); ^13^C NMR (CDCl_3_, 125 MHz) δ ppm: 55.9 (OCH_3_), 56.3 (OCH_3_), 106.4 (C), 112.1 (CH), 112.5 (CH), 114.9 (CN), 121.5 (C), 121.5 (CH), 126.5 (2CH), 127.3 (2CH), 127.4 (2CH), 128.1 (2CH), 132.9 (C), 138.5 (2C), 148.6 (=CH), 153.4 (C), 153.5 (2C), 162.5 (C=O); IR ν (cm^−1^): 2201, 1666, 1575, 1495, 1457, 1358, 1306, 1228, 1161, 1041, 945, 847, 812, 159, 701, 666. Anal. Calcd for C_24_H_18_N_2_O_3_S: C, 69.55; H, 4.38; N, 6.76. Found: C, 69.90; H, 4.56; N, 7.02%.

#### 3.2.7. (E)-3-(3-Chloro-4-methoxyphenyl)-2-(10H-phenothiazine-10-carbonyl)acrylonitrile (**1h**)

General procedure A was used with 3-oxo-3-(10*H*-phenothiazin-10-yl)propanenitrile **4a** (0.50 g, 1.87 mmol), 3-chloro-4-methoxybenzaldehyde **20** (0.38 g, 2.25 mmol), piperidine (3 drops), and glacial acetic acid (1 drop) in 15 mL acetonitrile to obtain pure **1h** (0.56 g, 1.35 mmol, 73% yield) as a yellow solid; mp > 250 °C (EtOH); ^1^H NMR (CDCl_3_, 500 MHz) δ ppm: 3.96 (s, 3H, OC*H*_3_), 6.93–6.99 (m, 1H, Ar*H*), 7.27–7.38 (m, 4H, Ar*H*), 7.51 (d, *J* = 7.5 Hz, 2H, Ar*H*), 7.61 (d, *J* = 7.5 Hz, 2H, Ar*H*), 7.81–7.86 (m, 2H, Ar*H*), 7.92 (s, 1H, =C*H*); ^13^C NMR (CDCl_3_, 125 MHz) δ ppm: 56.6 (OCH_3_), 105.5 (C), 112.2 (CH), 114.5 (CN), 123.5 (C), 125.7 (C), 126.4 (2CH), 127.4 (2CH), 127.5 (2CH), 128.3 (2CH), 130.8 (CH), 132.7 (CH), 132.9 (2C), 138.4 (2C), 152.3 (=CH), 158.5 (C), 162.2 (C=O); IR ν (cm^−1^): 2209, 2159, 1674, 1587, 1498, 1459, 1326, 1260, 1186, 1058, 1013, 915, 812, 757, 727, 690. Anal. Calcd for C_23_H_15_ClN_2_O_2_S: C, 65.95; H, 3.61; N, 6.69. Found: C, 66.23; H, 3.83; N, 6.92%.

#### 3.2.8. (E)-3-(3-Fluoro-4-methoxyphenyl)-2-(10H-phenothiazine-10-carbonyl)acrylonitrile (**1i**)

General procedure A was used with 3-oxo-3-(10*H*-phenothiazin-10-yl)propanenitrile **4a** (0.50 g, 1.87 mmol), 3-fluoro-4-methoxybenzaldehyde **21** (0.35 g, 2.25 mmol), piperidine (3 drops), and glacial acetic acid (3 drops) in 15 mL acetonitrile to obtain pure **1i** (0.52 g, 1.29 mmol, 69% yield) as a yellow solid; mp > 250 °C (EtOH); ^1^H NMR (CDCl_3_, 500 MHz) δ ppm: 3.95 (s, 3H, OC*H*_3_), 6.99 (t, *J* = 8.0 Hz, 1H, Ar*H*), 7.27–7.36 (m, 4H, Ar*H*), 7.50 (d, *J* = 7.5 Hz, 2H, Ar*H*), 7.61 (d, *J* = 7.5 Hz, 3H, Ar*H*), 7.67 (d, *J* = 12.0 Hz, 1H, Ar*H*), 7.94 (s, 1H, =C*H*); ^13^C NMR (CDCl_3_, 125 MHz) δ ppm: 56.5 (OCH_3_), 105.5 (C), 113.2 (d, *J* = 2.5 Hz, CH), 114.4 (CN), 117.6 (d, *J* = 18.75 Hz, CH), 125.3 (d, *J* = 7.5 Hz, C), 126.4 (2CH), 127.4 (2CH), 127.5 (2CH), 128.3 (2CH), 128.7 (d, *J* = 2.5 Hz, CH), 133.0 (2C), 138.4 (2C), 151.3 (d, *J* = 56.25 Hz, C), 152.4 (d, *J* = 180.0 Hz, C-F), 152.5 (d, *J* = 2.5 Hz, =CH), 162.2 (C=O); IR ν (cm^−1^): 2212, 2026, 1676, 1599, 1573, 1515, 1480, 1441, 1330, 1288, 1260, 1238, 1200, 1141, 1016, 973, 924, 871, 819, 761, 629. Anal. Calcd for C_23_H_15_FN_2_O_2_S: C, 68.64; H, 3.76; N, 6.96. Found: C, 68.90; H, 3.89; N, 7.13%.

#### 3.2.9. (E)-3-(4-Methoxy-3-nitrophenyl)-2-(10H-phenothiazine-10-carbonyl)acrylonitrile (**1j**)

General procedure A was used with 3-oxo-3-(10*H*-phenothiazin-10-yl)propanenitrile **4a** (0.50 g, 1.87 mmol), 4-methoxy-3-nitrobenzaldehyde **25** (0.39 g, 2.16 mmol), piperidine (3 drops), and glacial acetic acid (3 drops) in 15 mL acetonitrile to obtain pure **1j** (0.56 g, 1.31 mmol, 70% yield) as a yellow solid; mp > 250 °C (EtOH); ^1^H NMR (CDCl_3_, 500 MHz) δ ppm: 4.03 (s, 3H, OC*H*_3_), 7.15 (d, *J* = 8.5 Hz, 1H, Ar*H*), 7.28–7.37 (m, 4H, Ar*H*), 7.51 (dd, *J* = 8.0, 2.0 Hz, 2H, Ar*H*), 7.60 (d, *J* = 8.0 Hz, 2H, Ar*H*), 7.96 (s, 1H, =C*H*), 8.17–8.19 (m, 2H, Ar*H*); ^13^C NMR (CDCl_3_, 125 MHz) δ ppm: 57.1 (OCH_3_), 107.7 (C), 114.0 (CN), 114.2 (CH), 124.7 (C), 126.4 (2CH), 127.5 (2CH), 127.7 (2CH), 128.3 (2CH), 128.6 (CH), 133.0 (2C), 135.1 (CH), 138.2 (2C), 139.8 (C), 150.8 (=CH), 155.6 (C), 161.5 (C=O); IR ν (cm^−1^): 2207, 2160, 1666, 1595, 1532, 1460, 1330, 1283, 1220, 1184, 1082, 1003, 926, 865, 828, 761, 728, 666, 606. Anal. Calcd for C_23_H_15_N_3_O_4_S: C, 64.33; H, 3.52; N, 9.78. Found: C, 64.50; H, 3.72; N, 9.93%.

#### 3.2.10. (E)-3-(2,6-Dichlorophenyl)-2-(10H-phenothiazine-10-carbonyl)acrylonitrile (**1k**)

General procedure A was used with 3-oxo-3-(10*H*-phenothiazin-10-yl)propanenitrile **4a** (0.50 g, 1.87 mmol), 2,6-dichlorobenzaldehyde **26** (0.39 g, 2.22 mmol), piperidine (3 drops), and glacial acetic acid (3 drops) in 15 mL acetonitrile to obtain pure **1k** (0.56 g, 1.31 mmol, 70% yield) as a yellow solid; mp 237–239 °C (EtOH); ^1^H NMR (CDCl_3_, 500 MHz) δ ppm: 7.27–7.32 (m, 3H, Ar*H*), 7.33–7.39 (m, 4H, Ar*H*), 7.49 (d, *J* = 8.0 Hz, 2H, Ar*H*), 7.65–7.72 (m, 2H, Ar*H*), 8.06 (s, 1H, =C*H*); ^13^C NMR (CDCl_3_, 125 MHz) δ ppm: 112.2 (C), 118.1 (CN), 126.5 (2CH), 127.6 (2CH), 127.8 (2CH), 128.2 (2CH), 128.5 (2CH), 130.9 (2C), 131.5 (CH), 132.7 (C), 134.3 (2C), 137.9 (2C), 150.0 (=CH), 160.3 (C=O); IR ν (cm^−1^): 2159, 2032, 1663, 1618, 1578, 1479, 1459, 1430, 1343, 1264, 1186, 1031, 957, 821, 783, 768, 748, 726, 681. Anal. Calcd for C_22_H_12_ClN_2_OS: C, 62.42; H, 2.86; N, 6.62. Found: C, 62.78; H, 3.09; N, 6.83%.

#### 3.2.11. (E)-3-(3-Hydroxy-4-methoxyphenyl)-2-(10H-phenothiazine-10-carbonyl)acrylonitrile (**1l**)

General procedure A was used with 3-oxo-3-(10*H*-phenothiazin-10-yl)propanenitrile **4a** (0.30 g, 1.13 mmol), 3-hydroxy-4-methoxybenzaldehyde **23** (0.21 g, 1.38 mmol), piperidine (3 drops), and glacial acetic acid (3 drops) in 10 mL ethanol to obtain pure **1l** (0.25 g, 0.62 mmol, 55% yield) as a yellow solid; mp 218–220 °C (EtOH); ^1^H NMR (CDCl_3_, 400 MHz) δ ppm: 3.89 (s, 3H, OC*H*_3_), 6.15 (br s, 1H, O*H*), 6.96 (d, *J* = 8.0 Hz, 1H, Ar*H*), 7.21–7.38 (m, 5H, Ar*H*), 7.49 (d, *J* = 7.6 Hz, 2H, Ar*H*), 7.63 (d, *J* = 7.6 Hz, 2H, Ar*H*), 7.68 (d, *J* = 1.2 Hz, 1H, Ar*H*), 7.99 (s, 1H, =C*H*); ^13^C NMR (CDCl_3_, 100 MHz) δ ppm: 56.1 (OCH_3_), 103.2 (C), 110.7 (CH), 114.8 (CH), 115.1 (CN), 124.9 (C), 126.3 (2CH), 127.2 (2CH), 127.3 (2CH), 128.0 (2CH), 128.1 (CH), 132.8 (2C), 138.5 (2C), 146.7 (C), 150.2 (C), 154.3 (=CH), 162.5 (C=O). IR ν (cm^−1^): 3341, 2207, 1666, 1572, 1459, 1261, 757. Anal. Calcd for C_23_H_16_N_2_O_3_S: C, 68.98; H, 4.03; N, 7.00. Found: C, 68.79; H, 3.94; N, 6.87%.

#### 3.2.12. (E)-3-(1H-Indol-3-yl)-2-(10H-phenothiazine-10-carbonyl)acrylonitrile (**1n**)

General procedure A was used with 3-oxo-3-(10*H*-phenothiazin-10-yl)propanenitrile **4a** (0.50 g, 1.87 mmol), 1*H*-indole-3-carbaldehyde **30** (0.33 g, 2.27 mmol), piperidine (3 drops), and glacial acetic acid (3 drops) in 10 mL ethanol to obtain pure **1n** (0.63 g, 1.60 mmol, 85% yield) as a yellow solid; mp > 250 (EtOH); ^1^H NMR (CDCl_3_, 400 MHz) δ ppm: 7.27–7.35 (m, 6H, Ar*H*), 7.44 (td, *J* = 8.8, 4.0 Hz, 1H, Ar*H*), 7.50 (d, *J* = 7.6 Hz, 2H, Ar*H*), 7.68 (d, *J* = 7.6 Hz, 2H, Ar*H*), 7.80 (td, *J* = 8.8, 4.0 Hz, 1H, Ar*H*), 8.44 (d, *J* = 3.2 Hz, 1H, Ar*H*), 8.56 (s, 1H, =C*H*), 8.98 (br s, 1H, N*H*); ^13^C NMR (CDCl_3_, 100 MHz) δ ppm: 98.8 (C), 111.5 (C), 112.1 (CH), 116.7 (CN), 118.3 (CH), 122.5 (CH), 124.1 (CH), 126.5 (2CH), 127.1 (2CH), 127.2 (2CH), 127.4 (C), 128.0 (2CH), 129.7 (CH), 132.8 (2C), 135.5 (C), 138.8 (2C), 146.7 (=CH), 162.9 (C=O). IR ν (cm^−1^): 3293, 2216, 1651, 1561, 1459, 1329, 1291, 1227, 734. Anal. Calcd for C_24_H_15_N_3_OS: C, 73.26; H, 3.84; N, 10.68. Found: C, 73.38; H, 3.93; N, 10.82%.

#### 3.2.13. (E)-3-(5-Methoxy-1H-indol-3-yl)-2-(10H-phenothiazine-10-carbonyl)acrylonitrile (**1o**)

General procedure A was used with 3-oxo-3-(10*H*-phenothiazin-10-yl)propanenitrile **4a** (0.30 g, 1.13 mmol), 5-methoxy-1*H*-indole-3-carbaldehyde **31** (0.24 g, 1.37 mmol), piperidine (3 drops), and glacial acetic acid (1 drop) in 15 mL acetonitrile to obtain pure **1o** (0.33 g, 0.78 mmol, 61% yield) as a yellow solid; mp > 250 °C (EtOH); ^1^H NMR (CDCl_3_, 500 MHz) δ ppm: 3.91 (s, 3H, OC*H*_3_), 6.96 (dd, *J* = 8.5, 2.5 Hz, 1H, Ar*H*), 7.22 (d, *J* = 2.5 Hz, 1H, Ar*H*), 7.27–7.34 (m, 5H, Ar*H*), 7.50 (dd, *J* = 7.5, 1.5 Hz, 2H, Ar*H*), 7.68 (dd, *J* = 7.5, 1.5 Hz, 2H, Ar*H*), 8.38 (d, *J* = 3.0 Hz, 1H, Ar*H*), 8.54 (s, 1H, =C*H*), 8.89 (br s, 1H, N*H*); ^13^C NMR (CDCl_3_, 125 MHz) δ ppm: 56.0 (OCH_3_), 98.4 (C), 100.1 (CH), 111.7 (C), 113.0 (CH), 114.7 (CH), 116.8 (CN), 126.6 (2CH), 127.3 (2CH), 127.4 (2CH), 128.2 (2CH), 128.4 (C), 130.1 (CH), 130.4 (C), 133.0 (2C), 139.0 (2C), 146.8 (=CH), 156.4 (C), 163.2 (C=O); IR ν (cm^−1^): 3290, 2200, 1658, 1568, 1480, 1319, 1260, 1216, 1141, 1059, 1027, 927, 867, 803, 767, 729, 662, 632. Anal. Calcd for C_25_H_17_N_3_O_2_S: C, 70.90; H, 4.05; N, 9.92. Found: C, 71.29; H, 4.41; N, 9.62%.

#### 3.2.14. (E)-3-(5-Methoxy-1-methyl-1H-indol-3-yl)-2-(10H-phenothiazine-10-carbonyl)acrylonitrile (**1p**)

General procedure A was used with 3-oxo-3-(10*H*-phenothiazin-10-yl)propanenitrile **4a** (0.30 g, 1.13 mmol), 5-methoxy-1-methyl-1*H*-indole-3-carbaldehyde **32** (0.26 g, 1.37 mmol), piperidine (3 drops), and glacial acetic acid (1 drop) in 20 mL acetonitrile to obtain pure **1p** (0.35 g, 0.80 mmol, 63% yield) as a yellow solid; mp > 250 °C (EtOH); ^1^H NMR (CDCl_3_, 500 MHz) δ ppm: 3.80 (s, 3H, NC*H*_3_), 3.91 (s, 3H, OC*H*_3_), 6.97 (dd, *J* = 8.5, 2.5 Hz, 1H, Ar*H*), 7.22 (d, *J* = 2.5 Hz, 1H, Ar*H*), 7.22–7.28 (m, 3H, Ar*H*), 7.33 (td, *J* = 7.5, 1.0 Hz, 2H, Ar*H*), 7.49 (dd, *J* = 7.5, 1.0 Hz, 2H, Ar*H*), 7.68 (d, *J* = 7.5 Hz, 2H, Ar*H*), 8.26 (s, 1H, Ar*H*), 8.51 (s, 1H, =C*H*); ^13^C NMR (CDCl_3_, 125 MHz) δ ppm: 34.2 (CH_3_), 56.0 (OCH_3_), 96.7 (C), 100.2 (CH), 110.2 (C), 111.4 (CH), 114.2 (CH), 117.2 (CN), 126.6 (2CH), 127.1 (2CH), 127.3 (2CH), 128.1 (2CH), 129.4 (C), 131.8 (C), 133.0 (2C), 134.2 (CH), 139.1 (2C), 146.4 (=CH), 156.5 (C), 163.4 (C=O); IR ν (cm^−1^): 2205, 1667, 1629, 1575, 1517, 1459, 1395,1355, 1313, 1259, 1221, 1114, 1040, 836, 798, 764, 729, 702, 655, 609. Anal. Calcd for C_26_H_19_N_3_O_2_S: C, 71.38; H, 4.38; N, 9.60. Found: C, 71.26; H, 4.88; N, 9.73%.

#### 3.2.15. (E)-3-(1-Acetyl-1H-indol-3-yl)-2-(10H-phenothiazine-10-carbonyl)acrylonitrile (**1q**)

General procedure A was used with 3-oxo-3-(10*H*-phenothiazin-10-yl)propanenitrile **4a** (0.30 g, 1.13 mmol), 1-acetyl-1*H*-indole-3-carbaldehyde **33** (0.25 g, 1.35 mmol), piperidine (3 drops), and glacial acetic acid (1 drop) in 15 mL acetonitrile to obtain pure **1q** (0.35 g, 0.80 mmol, 71% yield) as a yellow solid; mp > 250 °C (EtOH); ^1^H NMR (CDCl_3_, 400 MHz) δ ppm: 2.65 (s, 3H, C*H*_3_), 7.28–7.38 (m, 4H, Ar*H*), 7.39–7.49 (m, 2H, Ar*H*), 7.52 (d, *J* = 7.6 Hz, 2H, Ar*H*), 7.66 (d, *J* = 7.6 Hz, 2H, Ar*H*), 7.72 (d, *J* = 7.6 Hz, 1H, Ar*H*), 8.44 (s, 1H, Ar*H*), 8.47 (d, *J* = 7.6 Hz, 1H, Ar*H*), 8.57 (s, 1H, =C*H*); ^13^C NMR (CDCl_3_, 100 MHz) δ ppm: 23.8 (CH_3_), 104.7 (C), 115.3 (C), 115.6 (CN), 117.0 (CH), 118.0 (CH), 124.9 (CH), 126.4 (2CH), 126.7 (CH), 127.3 (2CH), 127.5 (2CH), 128.1 (2CH), 128.4 (CH),128.9 (C), 132.8 (2C), 135.3 (C), 138.3 (2C), 144.4 (=CH), 161.5 (C=O), 168.8 (C=O). Anal. Calcd for C_26_H_17_N_3_O_2_S: C, 71.71; H, 3.93; N, 9.65. Found: C, 72.02; H, 4.06; N, 9.88%.

#### 3.2.16. (E)-3-(Anthracen-9-yl)-2-(10H-phenothiazine-10-carbonyl)acrylonitrile (**1r**)

General procedure A was used with 3-oxo-3-(10*H*-phenothiazin-10-yl)propanenitrile **4a** (0.50 g, 1.87 mmol), anthracene-9-carbaldehyde **34** (0.45 g, 2.18 mmol), piperidine (3 drops), and glacial acetic acid (3 drops) in 15 mL acetonitrile to obtain pure **1r** (0.72 g, 1.59 mmol, 85% yield) as a yellow solid; mp > 250 °C (EtOH); ^1^H NMR (CDCl_3_, 500 MHz) δ ppm: 7.35 (td, *J* = 8.0, 1.0 Hz, 2H, Ar*H*), 7.44 (td, *J* = 8.0, 1.0 Hz, 2H, Ar*H*), 7.47–7.51 (m, 4H, Ar*H*), 7.54 (dd, *J* = 7.5, 1.0 Hz, 2H, Ar*H*), 7.63–7.70 (m, 2H, Ar*H*), 7.82 (d, *J* = 7.5 Hz, 2H, Ar*H*), 7.98–8.03 (m, 2H, Ar*H*), 8.50 (s, 1H, =C*H*), 8.79 (s, 1H, Ar*H*); ^13^C NMR (CDCl_3_, 125 MHz) δ ppm: 113.4 (C), 117.2 (CN), 124.7 (2CH), 125.5 (C), 125.7 (2CH), 126.5 (2CH), 127.1 (2CH), 127.7 (4CH), 128.4 (2CH), 129.1 (2C), 129.2 (2CH), 130.2 (CH), 131.1 (3C), 132.9 (C), 138.4 (2C), 152.9 (=CH), 161.7 (C=O); IR ν (cm^−1^): 2227, 2156, 1673, 1598, 1459, 1330, 1263, 1200, 1127, 1076, 1030, 939, 885, 842, 793, 760, 729, 666, 629. Anal. Calcd for C_30_H_18_N_2_OS: C, 79.27; H, 3.99; N, 6.16. Found: C, 79.11; H, 3.75; N, 6.02%.

#### 3.2.17. (E)-3-(4-(Dimethylamino)phenyl)-2-(2-(methylthio)-10H-phenothiazine-10-carbonyl)acrylonitrile (**2b**)

General procedure A was used with 3-(2-(methylthio)-10*H*-phenothiazin-10-yl)-3-oxopropanenitrile **4b** (0.28 g, 0.90 mmol), 4-(dimethylamino)benzaldehyde **10** (0.16 g, 1.08 mmol), piperidine (3 drops), and glacial acetic acid (1 drop) in 5 mL acetonitrile to obtain pure **2b** (0.28 g, 1.58 mmol, 70% yield) as an orange solid; mp 221–223 °C (EtOH); ^1^H NMR (CDCl_3,_ 500 MHz) δ ppm: 2.46 (s, 3H, SC*H*_3_), 3.08 (s, 6H, 2C*H*_3_), 6.65 (d, *J* = 9.0 Hz, 2H, Ar*H*), 7.15 (dd, *J* = 8.0, 2.0 Hz, 1H, Ar*H*), 7.22–7.31 (m, 2H, Ar*H*), 7.35 (d, *J* = 8.0 Hz, 1H, Ar*H*), 7.47 (dd, *J* = 7.5, 1.5 Hz, 1H, Ar*H*), 7.56 (dd, *J* = 7.5, 1.5 Hz, 1H, Ar*H*), 7.64 (d, *J* = 2.0 Hz, 1H, Ar*H*), 7.81 (d, *J* = 9.0 Hz, 2H, Ar*H*), 7.98 (s, 1H, =C*H*); ^13^C NMR (CDCl_3_, 125 MHz) δ ppm: 16.5 (SCH_3_), 40.1 (2NCH_3_), 98.4 (C), 111.6 (2CH), 116.2 (C≡N), 120.1 (C), 124.7 (CH), 125.6 (CH), 126.4 (CH), 127.2 (CH), 127.3 (CH), 128.0 (CH), 128.2 (CH), 129.1 (C), 133.1 (C), 133.6 (2CH), 138.1 (C), 139.0 (C), 139.6 (C), 153.4 (C), 154.6 (=CH), 163.8 (C=O); IR ν (cm^−1^): 2200, 1664, 1571, 1526, 1313, 1182, 811, 751. Anal. Calcd for C_25_H_21_N_3_OS_2_: C, 67.69; H, 4.77; N, 9.47. Found: C, 67.90; H, 4.62; N, 9.33%.

#### 3.2.18. (E)-3-(4-(Dimethylamino)-2-methoxyphenyl)-2-(2-(methylthio)-10H-phenothiazine-10-carbonyl)acrylonitrile (**2g**)

General procedure A was used with 3-(2-(methylthio)-10*H*-phenothiazin-10-yl)-3-oxopropanenitrile **4b** (0.30 g, 0.96 mmol), 2-methoxy-4-(dimethylamino)benzaldehyde **22** (0.21 g, 1.16 mmol), piperidine (4 drops), and glacial acetic acid (1 drop) in 5 mL acetonitrile to obtain pure **2g** (0.37 g, 0.78 mmol, 82% yield) as a yellow solid; mp 184–186 °C (EtOH); ^1^H NMR (CDCl_3_, 500 MHz) δ ppm: 2.46 (s, 3H, SC*H*_3_), 3.08 (s, 6H, 2C*H*_3_), 3.88 (s, 3H, OC*H*_3_), 6.02 (d, *J* = 2.5 Hz, 1H, Ar*H*), 6.27 (dd, *J* = 9.0, 2.5 Hz, 1H, Ar*H*), 7.13 (dd, *J* = 8.5, 2.0 Hz, 1H, Ar*H*), 7.21–7.30 (m, 3H, Ar*H*), 7.33 (d, *J* = 7.5 Hz, 1H, Ar*H*), 7.58 (d, *J* = 7.5 Hz, 1H, Ar*H*), 7.67 (d, *J* = 2.0 Hz, 1H, Ar*H*), 8.20 (d, *J* = 9.0 Hz, 1H, ArH), 8.53 (s, 1H, =C*H*); ^13^C NMR (CDCl_3_, 125 MHz) δ ppm: 16.5 (SCH_3_), 40.3 (2NCH_3_), 55.5 (OCH_3_), 93.2 (CH), 96.9 (C), 105.1 (CH), 110.2 (C), 116.9 (C≡N), 124.7 (CH), 125.4 (CH), 126.4 (CH),127.0 (CH), 127.3 (CH), 127.9 (CH), 128.1 (CH), 129.1 (C), 130.5 (CH), 133.0 (C), 138.0 (C), 139.3 (C), 139.8 (C), 148.2 (=CH), 155.3 (C), 161.5 (C), 164.4 (C=O); IR ν (cm^−1^): 2204, 1669, 1560, 1461, 1247, 1122, 809, 748, 667. Anal. Calcd for C_26_H_23_N_3_O_2_S_2_: C, 65.94; H, 4.89; N, 8.87. Found: C, 66.23; H, 5.04; N, 9.12%.

#### 3.2.19. (E)-3-(1H-Indol-3-yl)-2-(2-(methylthio)-10H-phenothiazine-10-carbonyl)acrylonitrile (**2k**)

General procedure A was used with 3-(2-(methylthio)-10*H*-phenothiazin-10-yl)-3-oxopropanenitrile **4b** (0.28 g, 0.90 mmol), indole-3-carboxaldehyde **30** (0.16 g, 1.08 mmol), piperidine (3 drops), and glacial acetic acid (1 drop) in 6 mL ethanol to obtain pure **2k** (0.25 g, 0.57 mmol, 64% yield) as a yellowish solid; mp 194–195 °C (EtOH); ^1^H NMR (DMSO-*d*_6_, 500 MHz) δ ppm: 2.42 (s, 3H, SC*H*_3_), 7.20–7.30 (m, 3H, Ar*H*), 7.34 (td, *J* = 7.5, 1.5 Hz, 1H, Ar*H*), 7.39 (td, *J* = 7.5, 1.5 Hz, 1H, Ar*H*), 7.52 (d, *J* = 8.0 Hz, 1H, Ar*H*), 7.56 (dd, *J* = 7.5, 1.5 Hz, 1H, Ar*H*), 7.60 (dd, *J* = 7.5, 1.5 Hz, 1H, Ar*H*), 7.73 (dd, *J* = 8.0, 1.5 Hz, 2H, Ar*H*), 7.82 (dd, *J* = 7.5, 1.5 Hz, 1H, Ar*H*), 8.35 (s, 1H, Ar*H*), 8.42 (s, 1H, =C*H*), 12.40 (s, 1H, N*H*); ^13^C NMR (DMSO-*d*_6_, 125 MHz) δ ppm: 15.0 (SCH_3_), 96.5 (C), 110.1 (C), 112.9 (CH), 116.6 (C≡N), 118.2 (CH), 122.0 (CH), 123.5 (CH), 123.9 (CH), 124.8 (CH), 126.6 (CH), 126.9 (C), 127.3 (CH), 127.5 (CH), 127.7 (C), 128.0 (2CH), 130.9 (CH), 131.8 (C), 136.0 (C), 137.9 (C), 138.4 (C), 139.0 (C), 146.7 (=CH), 162.4 (C=O); IR ν (cm^−1^): 3331, 2214, 1650, 1562, 1329, 1140, 943, 735, 665. Anal. Calcd for C_25_H_17_N_3_OS_2_: C, 68.31; H, 3.90; N, 9.56. Found: C, 68.55; H, 3.73; N, 9.41%.

#### 3.2.20. (E)-3-(5-Methoxy-1H-indol-3-yl)-2-(2-(methylthio)-10H-phenothiazine-10-carbonyl)acrylonitrile (**2l**)

General procedure A was used with 3-(2-(methylthio)-10*H*-phenothiazin-10-yl)-3-oxopropanenitrile **4b** (0.30 g, 0.96 mmol), 5-methoxyindole-3-carboxaldehyde **31** (0.20 g, 1.14 mmol), piperidine (4 drops), and glacial acetic acid (1 drop) in 8 mL acetonitrile to obtain pure **2l** (0.32 g, 0.68 mmol, 70% yield) as a yellow solid; mp 218–220 °C (EtOH); ^1^H NMR (DMSO-*d*_6_, 500 MHz) δ ppm: 2.42 (s, 3H, SC*H*_3_), 3.84 (s, 3H, OC*H*_3_), 6.89 (dd, *J* = 8.5, 2.0 Hz, 1H, Ar*H*), 7.22 (dd, *J* = 8.0, 2.0 Hz, 1H, Ar*H*), 7.30–7.34 (m, 2H, Ar*H*), 7.38 (t, *J* = 7.5 Hz, 1H, Ar*H*), 7.44 (d, *J* = 8.5 Hz, 1H, Ar*H*), 7.51 (dd, *J* = 8.0, 2.0 Hz, 1H, Ar*H*), 7.59 (d, *J* = 7.5 Hz, 1H, Ar*H*), 7.70 (d, *J* = 7.5 Hz, 2H, Ar*H*), 8.29 (s, 1H, Ar*H*), 8.46 (s, 1H, =C*H*), 12.29 (s, 1H, N*H*); ^13^C NMR (DMSO-*d*_6_, 125 MHz) δ ppm: 15.0 (SCH_3_), 55.4 (OCH_3_), 95.6 (C), 100.1 (CH), 110.3 (C), 113.6 (CH), 113.7 (CH), 116.8 (C≡N), 123.8 (CH), 124.7 (CH), 126.5 (CH), 127.3 (CH), 127.5 (CH), 127.7 (C), 127.9 (C), 128.0 (2CH), 130.8 (C), 131.1 (CH), 131.8 (C), 137.9 (C), 138.5 (C), 139.1 (C), 147.1 (=CH), 155.6 (C), 162.6 (C=O); IR ν (cm^−1^): 3277, 2217, 1648, 1564, 1460, 1313, 1215, 1115, 929, 793, 739, 660. Anal. Calcd for C_26_H_19_N_3_O_2_S_2_: C, 66.50; H, 4.08; N, 8.95. Found: C, 66.72; H, 4.33; N, 8.87%.

#### 3.2.21. (E)-3-(5-Methoxy-1-methyl-1H-indol-3-yl)-2-(2-(methylthio)-10H-phenothiazine-10-carbonyl)acrylonitrile (**2m**)

General procedure A was used with 3-(2-(methylthio)-10*H*-phenothiazin-10-yl)-3-oxopropanenitrile **4b** (0.30 g, 0.96 mmol), 5-methoxy-1-methylindole-3-carboxaldehyde **32** (0.22 g, 1.16 mmol), piperidine (4 drops), and glacial acetic acid (1 drop) in 5 mL acetonitrile to obtain pure **2m** (0.36 g, 0.74 mmol, 79% yield) as a yellow solid; mp 235–237 °C (EtOH); ^1^H NMR (CDCl_3_, 500 MHz) δ ppm: 2.47 (s, 3H, SC*H*_3_), 3.82 (s, 3H, NC*H*_3_), 3.91 (s, 3H, OC*H*_3_), 6.98 (dd, *J* = 8.5, 2.0 Hz, 1H, Ar*H*), 7.16 (dd, *J* = 8.5, 2.0 Hz, 1H, Ar*H*), 7.23 (d, *J* = 2.0 Hz, 1H, Ar*H*), 7.25–7.33 (m, 3H, Ar*H*), 7.37 (d, *J* = 8.5 Hz, 1H, Ar*H*), 7.49 (dd, *J* = 7.5, 2.0 Hz, 1H, Ar*H*), 7.59 (dd, *J* = 7.5, 2.0 Hz, 1H, Ar*H*), 7.69 (d, *J* = 2.0 Hz, 1H, Ar*H*), 8.28 (s, 1H, Ar*H*), 8.53 (s, 1H, =C*H*); ^13^C NMR (125 MHz, CDCl_3_) δ ppm: 16.4 (SCH_3_), 34.3 (NCH_3_), 56.0 (OCH_3_), 96.5 (C), 100.1 (CH), 110.2 (C), 111.4 (CH), 114.3 (CH), 117.2 (C≡N), 124.7 (CH), 125.4 (CH), 126.5 (CH), 127.3 (CH), 127.4 (CH), 128.0 (CH), 128.2 (CH), 129.0 (C), 129.4 (C), 131.8 (C), 133.2 (C), 134.2 (CH), 138.2 (C), 139.1 (C), 139.6 (C), 146.5 (=CH), 156.5 (C), 163.4 (C=O); IR ν (cm^−1^): 3117, 2203, 1647, 1566, 1460, 1301, 1223, 1115, 1076, 804, 733, 642. Anal. Calcd for C_27_H_21_N_3_O_2_S_2_: C, 67.06; H, 4.38; N, 8.69. Found: C, 67.35; H, 4.24; N, 8.75%.

#### 3.2.22. (E)-3-(2-Methoxynaphthalen-1-yl)-2-(2-(methylthio)-10H-phenothiazine-10-carbonyl)acrylonitrile (**2n**)

General procedure A was used with 3-(2-(methylthio)-10*H*-phenothiazin-10-yl)-3-oxopropanenitrile **4b** (0.30 g, 0.96 mmol), 2-methoxy-1-naphthaldehyde **35** (0.21 g, 1.16 mmol), piperidine (4 drops), and glacial acetic acid (1 drop) in 6 mL acetonitrile to obtain pure **2n** (0.28 g, 0.58 mmol, 61% yield) as a yellowish-green solid; mp 208–211 °C (EtOH); ^1^H NMR (CDCl_3_, 500 MHz) δ ppm: 2.50 (s, 3H, SC*H*_3_), 3.96 (s, 3H, OC*H*_3_), 7.19 (dd, *J* = 8.5, 2.0 Hz, 1H, Ar*H*), 7.26–7.34 (m, 2H, Ar*H*), 7.36–7.41 (m, 3H, Ar*H*), 7.51 (t, *J* = 8.0 Hz, 2H, Ar*H*), 7.58 (d, *J* = 8.5 Hz, 1H, Ar*H*), 7.62 (s, 1H, Ar*H*), 7.75 (d, *J* = 7.5 Hz, 1H, Ar*H*), 7.79 (d, *J* = 8.5 Hz, 1H, Ar*H*), 7.92 (d, *J* = 8.5 Hz, 1H, Ar*H*), 8.51 (s, 1H, =C*H*); ^13^C NMR (CDCl_3_, 125 MHz) δ ppm: 16.0 (SCH_3_), 56.0 (OCH_3_), 113.0 (CH), 114.1 (C), 114.6 (C), 123.3 (CH), 124.2 (CH), 124.5 (CH), 125.5 (CH), 126.7 (CH), 127.3 (CH), 127.4 (CH), 128.0 (CH), 128.2 (2CH), 128.5 (2C), 128.8 (CH), 128.9 (C), 132.1 (C), 132.9 (C), 133.4 (CH), 138.4 (C), 138.6 (C), 139.1 (C), 149.9 (=CH), 156.0 (C), 162.3 (C=O); IR ν (cm^−1^): 2218, 1662, 1462, 1316, 1258, 1152, 1088, 939, 812, 745. Anal. Calcd for C_28_H_20_N_2_O_2_S_2_: C, 69.97; H, 4.19; N, 5.83. Found: C, 70.23; H, 4.35; N, 5.94%.

#### 3.2.23. (E)-2-(9H-Carbazole-9-carbonyl)-3-(4-(dimethylamino)phenyl)acrylonitrile (**3a**)

General procedure A was used with 3-(9*H*-carbazol-9-yl)-3-oxopropanenitrile **5** (0.40 g, 1.70 mmol), 4-(dimethylamino)benzaldehyde **10** (0.31 g, 2.05 mmol), piperidine (3 drops), and glacial acetic acid (1 drop) in 15 mL acetonitrile to obtain pure **3a** (0.54 g, 1.48 mmol, 87% yield) as a white solid; mp 228–230 °C (EtOH); ^1^H NMR (CDCl_3_, 500 MHz) δ ppm: 3.11 (s, 6H, 2C*H*_3_), 6.71 (d, *J* = 9.0 Hz, 2H, Ar*H*), 7.36 (t, *J* = 7.5 Hz, 2H, Ar*H*), 7.44 (t, *J* = 7.5 Hz, 2H, Ar*H*), 7.91–8.04 (m, 7H, =C*H*+6Ar*H*); ^13^C NMR (CDCl_3_, 125 MHz) δ ppm: 40.2 (2CH_3_), 98.3 (C), 111.8 (2CH), 115.4 (2CH), 117.7 (CN), 119.7 (C), 120.2 (2CH), 123.5 (2CH), 126.0 (2C), 127.0 (2CH), 134.5 (2CH), 138.7 (2C), 154.1 (C), 154.5 (=CH), 164.8 (C=O); IR ν (cm^−1^): 2210, 1651, 1608, 1556, 1515, 1435, 1383, 1296, 1169, 1070, 939, 889, 822, 760, 682, 617. Anal. Calcd for C_24_H_19_N_3_O: C, 78.88; H, 5.24; N, 11.50. Found: C, 78.66; H, 5.09; N, 11.37%.

#### 3.2.24. (E)-2-(9H-Carbazole-9-carbonyl)-3-(4-methoxyphenyl)acrylonitrile (**3b**)

General procedure A was used with 3-(9*H*-carbazol-9-yl)-3-oxopropanenitrile **5** (0.40 g, 1.70 mmol), 4-methoxybenzaldehyde **11** (0.28 g, 2.04 mmol), piperidine (4 drops), and glacial acetic acid (1 drop) in 15 mL acetonitrile to obtain pure **3b** (0.44 g, 1.25 mmol, 74% yield) as a yellow solid; mp 173–175 °C (EtOH); ^1^H NMR (CDCl_3_, 500 MHz) δ ppm: 3.92 (s, 3H, OC*H*_3_), 7.04 (d, *J* = 7.0 Hz, 2H, Ar*H*), 7.40 (td, *J* = 7.5, 1.0 Hz, 2H, Ar*H*), 7.46 (td, *J* = 7.5, 1.0 Hz, 2H, Ar*H*), 7.96 (d, *J* = 7.5 Hz, 2H, Ar*H*), 7.97 (s, 1H, =C*H*), 8.01 (d, *J* = 7.5 Hz, 2H, Ar*H*), 8.05 (d, *J* = 7.0 Hz, 2H, Ar*H*); ^13^C NMR (CDCl_3_, 125 MHz) δ ppm: 55.9 (OCH_3_), 104.0 (C), 115.1 (2CH), 115.5 (2CH), 116.2 (CN), 120.3 (2CH), 124.1 (2CH), 124.7 (C), 126.3 (2C), 127.3 (2CH), 133.8 (2CH), 138.5 (2C), 154.7 (=CH), 163.4 (C), 164.3 (C=O); IR ν (cm^−1^): 2208, 2168, 1670, 1577, 1512, 1442, 1311, 1271, 1217, 1177, 1070, 1025, 977, 893, 839, 751, 687, 617. Anal. Calcd for C_23_H_16_N_2_O_2_: C, 78.39; H, 4.58; N, 7.95. Found: C, 78.47; H, 4.68; N, 8.11%.

#### 3.2.25. (E)-3-(4-Bromophenyl)-2-(9H-carbazole-9-carbonyl)acrylonitrile (**3c**)

General procedure A was used with 3-(9*H*-carbazol-9-yl)-3-oxopropanenitrile **5** (0.40 g, 1.70 mmol), 4-bromobenzaldehyde **12** (0.38 g, 2.04 mmol), piperidine (3 drops), and glacial acetic acid (1 drop) in 15 mL acetonitrile to obtain pure **3c** (0.52 g, 1.30 mmol, 76% yield) as a yellow solid; mp 193–194 °C (EtOH); ^1^H NMR (CDCl_3_, 500 MHz) δ ppm: 7.42 (t, *J* = 7.5 Hz, 2H, Ar*H*), 7.47 (t, *J* = 7.5 Hz, 2H, Ar*H*), 7.68 (d, *J* = 7.5 Hz, 2H, Ar*H*), 7.89 (d, *J* = 7.5 Hz, 2H, Ar*H*), 7.92 (s, 1H, =C*H*), 7.95 (d, *J* = 7.5 Hz, 2H, Ar*H*), 8.01 (d, *J* = 7.5 Hz, 2H, Ar*H*); ^13^C NMR (CDCl_3_, 125 MHz) δ ppm: 108.3 (C), 115.2 (CN), 115.6 (2CH), 120.4 (2CH), 124.5 (2CH), 126.5 (2C), 127.4 (2CH), 128.7 (C), 130.6 (C), 132.2 (2CH), 133.0 (2CH), 138.3 (2C), 153.3 (=CH), 162.3 (C=O); IR ν (cm^−1^): 2211, 1654, 1579, 1478, 1443, 1404, 1369, 1332, 1185, 1119, 1067, 1006, 953, 916, 822, 748, 692, 623. Anal. Calcd for C_22_H_13_BrN_2_O: C, 65.85; H, 3.27; N, 6.98. Found: C, 65.93; H, 3.31; N, 7.06%.

#### 3.2.26. (E)-2-(9H-Carbazole-9-carbonyl)-3-(3-hydroxyphenyl)acrylonitrile (**3d**)

General procedure A was used with 3-(9*H*-carbazol-9-yl)-3-oxopropanenitrile **5** (0.40 g, 1.70 mmol), 3-hydroxybenzaldehyde **15** (0.25 g, 2.05 mmol), piperidine (3 drops), and glacial acetic acid (1 drop) in 15 mL acetonitrile to obtain pure **3d** (0.28 g, 0.82 mmol, 48% yield) as a yellow solid; mp 202–204 °C (EtOH); ^1^H NMR (DMSO-*d*_6_, 500 MHz) δ ppm: 7.10 (s, 1H, Ar*H*), 7.21–7.65 (m, 7H, Ar*H*), 7.80–8.45 (m, 5H, =C*H*+4Ar*H*), 10.06 (br s, 1H, O*H*); ^13^C NMR (DMSO-*d*_6_, 125 MHz) δ ppm: 106.1 (C), 115.4 (2CH), 115.7 (CN), 116.3 (CH), 120.6 (2CH), 120.9 (CH), 122.2 (CH), 124.1 (2CH), 125.6 (2C), 127.3 (2CH), 130.6 (CH), 132.9 (C), 137.9 (2C), 155.8 (=CH), 157.9 (C), 162.7 (C=O); IR ν (cm^−1^): 3346, 2226, 1666, 1583, 1442, 1359, 1329, 1300, 1275, 1214, 1177, 984, 957, 867, 750, 683, 627. Anal. Calcd for C_22_H_14_N_2_O_2_: C, 78.09; H, 4.17; N, 8.28. Found: C, 78.40; H, 4.36; N, 8.51%.

#### 3.2.27. (E)-2-(9H-Carbazole-9-carbonyl)-3-(4-nitrophenyl)acrylonitrile (**3e**)

General procedure A was used with 3-(9*H*-carbazol-9-yl)-3-oxopropanenitrile **5** (0.30 g, 1.28 mmol), 4-nitrobenzaldehyde **16** (0.23 g, 1.52 mmol), piperidine (3 drops), and glacial acetic acid (1 drop) in 10 mL acetonitrile to obtain pure **3e** (0.31 g, 0.84 mmol, 66% yield) as an orange solid; mp 221–223 °C (EtOH); ^1^H NMR (CDCl_3_, 500 MHz) δ ppm: 7.41–7.51 (m, 4H, Ar*H*), 7.96 (d, *J* = 8.0 Hz, 2H, Ar*H*), 8.01 (s, 1H, =C*H*), 8.03 (d, *J* = 8.0 Hz, 2H, Ar*H*), 8.17 (d, *J* = 8.5 Hz, 2H, Ar*H*), 8.38 (d, *J* = 8.5 Hz, 2H, Ar*H*); ^13^C NMR (CDCl_3_, 125 MHz) δ ppm: 112.0 (C), 114.5 (CN), 115.7 (2CH), 120.6 (2CH), 124.7 (2CH), 124.9 (2CH), 126.7 (2C), 127.6 (2CH), 131.4 (2CH), 137.2 (C), 138.1 (2C), 150.0 (C), 151.0 (=CH), 161.3 (C=O); IR ν (cm^−1^): 2210, 1688, 1589, 1519, 1437, 1329, 1301, 1215, 1182, 1108, 1034, 1009, 937, 849, 798, 745, 654, 615. Anal. Calcd for C_22_H_13_N_3_O_3_: C, 71.93; H, 3.57; N, 11.44. Found: C, 71.77; H, 3.42; N, 11.49%.

#### 3.2.28. (E)-2-(9H-Carbazole-9-carbonyl)-3-(3,4-dimethoxyphenyl)acrylonitrile (**3f**)

General procedure A was used with 3-(9*H*-carbazol-9-yl)-3-oxopropanenitrile **5** (0.40 g, 1.71 mmol), 3,4-dimethoxybenzaldehyde **18** (0.34 g, 2.04 mmol), piperidine (3 drops), and glacial acetic acid (1 drop) in 15 mL acetonitrile to obtain pure **3f** (0.49 g, 1.28 mmol, 76% yield) as a yellow solid; mp 191–193 °C (EtOH); ^1^H NMR (CDCl_3_, 500 MHz) δ ppm: 3.98 (s, 3H, OC*H*_3_), 4.00 (s, 3H, OC*H*_3_), 6.98 (d, *J* = 8.5 Hz, 1H, Ar*H*), 7.41 (t, *J* = 7.5 Hz, 2H, Ar*H*), 7.45–7.51 (m, 3H, Ar*H*), 7.88 (d, *J* = 1.5 Hz, 1H, Ar*H*), 7.95 (s, 1H, =C*H*), 7.97 (d, *J* = 8.0 Hz, 2H, Ar*H*), 8.03 (d, *J* = 8.0 Hz, 2H, Ar*H*); ^13^C NMR (CDCl_3_, 125 MHz) δ ppm: 56.3 (OCH_3_), 56.4 (OCH_3_), 104.0 (C), 111.3 (CH), 111.7 (CH), 115.5 (2CH), 116.4 (CN), 120.3 (2CH), 124.2 (2CH), 125.0 (C), 126.3 (2C), 127.3 (2CH), 128.2 (CH), 138.5 (2C), 149.7 (C), 154.2 (C), 155.0 (=CH), 163.4 (C=O); IR ν (cm^−1^): 2849, 2208, 1672, 1586, 1510, 1441, 1357, 1327, 1274, 1217, 1164, 1071, 1017, 966, 853, 814, 779, 748, 690, 628. Anal. Calcd for C_24_H_18_N_2_O_3_: C, 75.38; H, 4.74; N, 7.33. Found: C, 75.80; H, 4.91; N, 7.56%.

#### 3.2.29. (E)-2-(9H-Carbazole-9-carbonyl)-3-(2,5-dimethoxyphenyl)acrylonitrile (**3g**)

General procedure A was used with 3-(9*H*-carbazol-9-yl)-3-oxopropanenitrile **5** (0.36 g, 1.54 mmol), 2,5-dimethoxybenzaldehyde **19** (0.31 g, 1.87 mmol), piperidine (3 drops), and glacial acetic acid (1 drop) in 15 mL acetonitrile to obtain pure **3g** (0.35 g, 0.92 mmol, 60% yield) as a yellow solid; mp 156–158 °C (EtOH); ^1^H NMR (CDCl_3_, 500 MHz) δ ppm: 3.81 (s, 3H, OC*H*_3_), 3.87 (s, 3H, OC*H*_3_), 6.93 (d, *J* = 9.0 Hz, 1H, Ar*H*), 7.15 (dd, *J* = 9.0, 2.0 Hz, 1H, Ar*H*), 7.41 (t, *J* = 7.5 Hz, 2H, Ar*H*), 7.48 (t, *J* = 7.5 Hz, 2H, Ar*H*), 7.96–8.05 (m, 5H, =C*H*+4Ar*H*), 8.50 (s, 1H, Ar*H*); ^13^C NMR (CDCl_3_, 125 MHz) δ ppm: 56.1 (OCH_3_), 56.3 (OCH_3_), 106.9 (C), 112.3 (CH), 112.9 (CH), 115.8 (2CH), 115.9 (CN), 120.3 (2CH), 121.1 (C), 122.7 (CH), 124.2 (2CH), 126.4 (2C), 127.3 (2CH), 138.5 (2C), 149.4 (=CH), 153.7 (C), 154.2 (C), 163.2 (C=O); IR ν (cm^−1^): 2214, 1671, 1496, 1444, 1361, 1333, 1302, 1259, 1232, 1186, 1075, 1039, 944, 823, 749, 616. Anal. Calcd for C_24_H_18_N_2_O_3_: C, 75.38; H, 4.74; N, 7.33. Found: C, 75.75; H, 4.88; N, 7.51%.

#### 3.2.30. (E)-2-(9H-Carbazole-9-carbonyl)-3-(3-chloro-4-methoxyphenyl)acrylonitrile (**3h**)

General procedure A was used with 3-(9*H*-carbazol-9-yl)-3-oxopropanenitrile **5** (0.30 g, 1.28 mmol), 3-chloro-4-methoxybenzaldehyde **20** (0.26 g, 1.52 mmol), piperidine (3 drops), and glacial acetic acid (1 drop) in 10 mL acetonitrile to obtain pure **3h** (0.42 g, 1.09 mmol, 84% yield) as a yellow solid; mp 196–198 °C (EtOH); ^1^H NMR (CDCl_3_, 500 MHz) δ ppm: 4.00 (s, 3H, OC*H*_3_), 7.05 (d, *J* = 8.0 Hz, 1H, Ar*H*), 7.40 (t, *J* = 7.5 Hz, 2H, Ar*H*), 7.46 (t, *J* = 7.5 Hz, 2H, Ar*H*), 7.87 (s, 1H, Ar*H*), 7.94 (d, *J* = 8.0 Hz, 2H, Ar*H*), 7.99–8.06 (m, 4H, =C*H*+3Ar*H*); ^13^C NMR (CDCl_3_, 125 MHz) δ ppm: 56.7 (OCH_3_), 105.7 (C), 112.4 (CH), 115.5 (2CH), 115.6 (CN), 120.4 (2CH), 124.0 (C), 124.3 (2CH), 125.2 (C), 126.4 (2C), 127.3 (2CH), 131.6 (CH), 133.1 (CH), 138.4 (2C), 153.0 (=CH), 159.3 (C), 162.8 (C=O); IR ν (cm^−1^): 2205, 1665, 1570, 1502, 1443, 1319, 1273, 1183, 1067, 1017, 955, 882, 846, 809, 760, 722, 676, 618. Anal. Calcd for C_23_H_15_ClN_2_O_2_: C, 71.41; H, 3.91; N, 7.24. Found: C, 71.29; H, 3.78; N, 7.10%.

#### 3.2.31. (E)-2-(9H-Carbazole-9-carbonyl)-3-(3-fluoro-4-methoxyphenyl)acrylonitrile (**3i**)

General procedure A was used with 3-(9*H*-carbazol-9-yl)-3-oxopropanenitrile **5** (0.30 g, 1.28 mmol), 3-fluoro-4-methoxybenzaldehyde **21** (0.24 g, 1.56 mmol), piperidine (3 drops), and glacial acetic acid (1 drop) in 15 mL acetonitrile to obtain pure **3i** (0.27 g, 0.72 mmol, 56% yield) as a yellow solid; mp 214–216 °C (EtOH); ^1^H NMR (CDCl_3_, 500 MHz) δ ppm: 4.00 (s, 3H, OC*H*_3_), 7.09 (t, *J* = 8.5 Hz, 1H, Ar*H*), 7.41 (t, *J* = 7.5 Hz, 2H, Ar*H*), 7.47 (t, *J* = 7.5 Hz, 2H, Ar*H*), 7.79 (d, *J* = 8.5 Hz, 1H, Ar*H*), 7.88–7.92 (m, 2H, =C*H*+Ar*H*), 7.95 (d, *J* = 7.5 Hz, 2H, Ar*H*), 8.02 (d, *J* = 7.5 Hz, 2H, Ar*H*); ^13^C NMR (CDCl_3_, 125 MHz) δ ppm: 56.6 (OCH_3_), 105.7 (C), 113.5 (d, *J* = 2.5 Hz, CH), 115.6 (2CH), 115.7 (CN), 118.0 (d, *J* = 18.75 Hz, CH), 120.4 (2CH), 124.3 (2CH), 124.8 (d, *J* = 6.25 Hz, C), 126.4 (2C), 127.3 (2CH), 129.5 (d, *J* = 3.75 Hz, CH), 138.4 (2C), 151.9 (d, *J* = 145 Hz, C-F), 152.9 (d, *J* = 92.5 Hz, C), 153.4 (d, *J* = 2.5 Hz, =CH), 163.9 (C=O); IR ν (cm^−1^): 2208, 1668, 1562, 1518, 1445, 1327, 1283, 1187, 1128, 1019, 961, 875, 808, 760, 685, 621. Anal. Calcd for C_23_H_15_FN_2_O_2_: C, 74.59; H, 4.08; N, 7.56. Found: C, 74.77; H, 4.32; N, 7.74%.

#### 3.2.32. (E)-2-(9H-Carbazole-9-carbonyl)-3-(3-hydroxy-4-methoxyphenyl)acrylonitrile (**3j**)

General procedure A was used with 3-(9*H*-carbazol-9-yl)-3-oxopropanenitrile **5** (0.40 g, 1.70 mmol), 3-hydroxy-4-methoxybenzaldehyde **23** (0.31 g, 2.04 mmol), piperidine (3 drops), and glacial acetic acid (1 drop) in 15 mL acetonitrile to obtain pure **3j** (0.51 g, 1.39 mmol, 81% yield) as a yellow solid; mp 211–213 °C (EtOH); ^1^H NMR (CDCl_3_, 500 MHz) δ ppm: 4.01 (s, 3H, OC*H*_3_), 5.76 (br s, 1H, O*H*), 6.99 (d, *J* = 8.5 Hz, 1H, Ar*H*), 7.41 (td, *J* = 7.5, 1.0 Hz, 2H, Ar*H*), 7.47 (td, *J* = 7.5, 1.0 Hz, 2H, Ar*H*), 7.62 (dd, *J* = 8.5, 2.0 Hz, 1H, Ar*H*), 7.70 (d, *J* = 2.0 Hz, 1H, Ar*H*), 7.91 (s, 1H, =C*H*), 7.96 (d, *J* = 7.5 Hz, 2H, Ar*H*), 8.02 (d, *J* = 7.5 Hz, 2H, Ar*H*); ^13^C NMR (CDCl_3_, 125 MHz) δ ppm: 56.4 (OCH_3_), 104.8 (C), 111.0 (CH), 115.6 (2CH), 115.9 (CN), 116.4 (CH), 120.3 (2CH), 124.2 (2CH), 125.5 (C), 125.9 (CH), 126.4 (2C), 127.3 (2CH), 138.5 (2C), 146.3 (C), 151.4 (C), 154.8 (=CH), 163.4 (C=O); IR ν (cm^−1^): 3325, 2208, 1641, 1590, 1509, 1440, 1370, 1331, 1277, 1209, 1147, 1072, 1022, 980, 933, 872, 816, 779, 742, 714. Anal. Calcd for C_23_H_16_N_2_O_3_: C, 74.99; H, 4.38; N, 7.60. Found: C, 75.26; H, 4.49; N, 7.78%.

#### 3.2.33. (E)-2-(9H-Carbazole-9-carbonyl)-3-(2,6-dichlorophenyl)acrylonitrile (**3k**)

General procedure A was used with 3-(9*H*-carbazol-9-yl)-3-oxopropanenitrile **5** (0.40 g, 1.70 mmol), 2,6-dichlorobenzaldehyde **26** (0.36 g, 2.05 mmol), piperidine (3 drops), and glacial acetic acid (1 drop) in 10 mL acetonitrile to obtain pure **3k** (0.48 g, 1.23 mmol, 72% yield) as a yellow solid; mp 173–175 °C (EtOH); ^1^H NMR (CDCl_3_, 500 MHz) δ ppm: 7.10–7.60 (m, 8H, Ar*H*), 7.90–8.25 (m, 4H, =C*H*+3Ar*H*); ^13^C NMR (CDCl_3_, 125 MHz) δ ppm: 110.7 (CH), 113.3 (C), 115.9 (CH), 118.7 (CN), 119.6 (CH), 120.4 (2CH), 123.5 (C), 124.8 (CH), 125.9 (CH), 126.6 (C), 127.4 (CH), 128.8 (CH), 130.4 (C), 132.0 (2CH), 134.4 (C), 138.1 (2C), 139.6 (C), 150.8 (=CH), 160.4 (C=O); IR ν (cm^−1^): 2158, 1679, 1599, 1449, 1365, 1325, 1213, 1187, 1093, 995, 928, 859, 788, 721, 678. Anal. Calcd for C_22_H_12_Cl_2_N_2_O: C, 67.54; H, 3.09; N, 7.16. Found: C, 67.31; H, 3.02; N, 7.04%.

#### 3.2.34. (E)-2-(9H-Carbazole-9-carbonyl)-3-(4-hydroxy-3-methoxyphenyl)acrylonitrile (**3l**)

General procedure A was used with 3-(9*H*-carbazol-9-yl)-3-oxopropanenitrile **5** (0.40 g, 1.70 mmol), 4-hydroxy-3-methoxybenzaldehyde **24** (0.31 g, 2.04 mmol), piperidine (3 drops), and glacial acetic acid (1 drop) in 15 mL acetonitrile to obtain pure **3l** (0.48 g, 1.30 mmol, 77% yield) as a yellow solid; mp 212–214 °C (EtOH); ^1^H NMR (DMSO-*d*_6_, 500 MHz) δ ppm: 3.90 (s, 3H, OC*H*_3_), 7.16 (d, *J* = 8.5 Hz, 1H, Ar*H*), 7.45 (t, *J* = 7.5 Hz, 2H, Ar*H*), 7.48–7.55 (m, 3H, Ar*H*), 7.71 (d, *J* = 2.0 Hz, 1H, Ar*H*), 7.93 (d, *J* = 8.5 Hz, 2H, Ar*H*), 8.16 (s, 1H, =C*H*), 8.24 (d, *J* = 7.5 Hz, 2H, Ar*H*), 9.77 (br s, 1H, O*H*); ^13^C NMR (DMSO-*d*_6_, 125 MHz) δ ppm: 55.9 (OCH_3_), 102.0 (C), 112.2 (CH), 115.2 (2CH), 115.8 (CH), 116.3 (CN), 120.5 (2CH), 123.8 (2CH), 124.5 (C), 125.4 (2C), 126.4 (CH), 127.2 (2CH), 137.9 (2C), 147.0 (C), 153.2 (C), 155.9 (=CH), 163.4 (C=O); IR ν (cm^−1^): 3322, 2208, 1641, 1590, 1509, 1440, 1370, 1332, 1277, 1209, 1148, 1072, 1023, 980, 933, 872, 816, 779, 742, 714, 626. Anal. Calcd for C_23_H_16_N_2_O_3_: C, 74.99; H, 4.38; N, 7.60. Found: C, 75.27; H, 4.61; N, 7.82%.

#### 3.2.35. (E)-2-(9H-Carbazole-9-carbonyl)-3-(4-methoxy-3-nitrophenyl)acrylonitrile (**3m**)

General procedure A was used with 3-(9*H*-carbazol-9-yl)-3-oxopropanenitrile **5** (0.30 g, 1.28 mmol), 3-nitro-4-methoxybenzaldehyde **25** (0.28 g, 1.54 mmol), piperidine (3 drops), and glacial acetic acid (1 drop) in 10 mL acetonitrile to obtain pure **3m** (0.35 g, 0.88 mmol, 68% yield) as a yellow solid; mp 233–235 °C (EtOH); ^1^H NMR (CDCl_3_, 500 MHz) δ ppm: 4.10 (s, 3H, OC*H*_3_), 7.28 (d, *J* = 8.5 Hz, 1H, Ar*H*), 7.43 (t, *J* = 7.5 Hz, 2H, Ar*H*), 7.49 (t, *J* = 7.5 Hz, 2H, Ar*H*), 7.93–7.96 (m, 3H, =C*H*+2Ar*H*), 8.03 (d, *J* = 7.5 Hz, 2H, Ar*H*), 8.38 (d, *J* = 2.0 Hz, 1H, Ar*H*), 8.44 (dd, *J* = 8.5, 2.0 Hz, 1H, Ar*H*); ^13^C NMR (CDCl_3_, 125 MHz) δ ppm: 57.3 (OCH_3_), 107.9 (C), 114.6 (CH), 115.2 (CN), 115.6 (2CH), 120.5 (2CH), 124.3 (C), 124.6 (2CH), 126.6 (2C), 127.5 (2CH), 129.1 (CH), 135.6 (CH), 138.3 (2C), 140.1 (C), 151.5 (=CH), 156.3 (C), 162.1 (C=O); IR ν (cm^−1^): 2208, 1665, 1617, 1578, 1534, 1501, 1444, 1357, 1330, 1294, 1167, 1090, 1011, 959, 896, 862, 819, 762, 665, 622. Anal. Calcd for C_23_H_15_N_3_O_4_: C, 69.52; H, 3.80; N, 10.57. Found: C, 69.73; H, 3.95; N, 10.89%.

#### 3.2.36. (E)-2-(9H-Carbazole-9-carbonyl)-3-(3,4,5-trimethoxyphenyl)acrylonitrile (**3n**)

General procedure A was used with 3-(9*H*-carbazol-9-yl)-3-oxopropanenitrile **5** (0.40 g, 1.70 mmol), 3,4,5-trimethoxybenzaldehyde **27** (0.40 g, 2.05 mmol), piperidine (3 drops), and glacial acetic acid (3 drops) in 15 mL acetonitrile to obtain pure **3n** (0.28 g, 0.68 mmol, 40% yield) as a yellow solid; mp 178–180 °C (EtOH); ^1^H NMR (CDCl_3_, 500 MHz) δ ppm: 3.93 (s, 6H, 2OC*H*_3_), 3.99 (s, 3H, OC*H*_3_), 7.34 (s, 2H, Ar*H*), 7.42 (t, *J* = 7.5 Hz, 2H, Ar*H*), 7.48 (t, *J* = 7.5 Hz, 2H, Ar*H*), 7.91 (s, 1H, =C*H*), 7.97 (d, *J* = 7.5 Hz, 2H, Ar*H*), 8.03 (d, *J* = 7.5 Hz, 2H, Ar*H*); ^13^C NMR (CDCl_3_, 125 MHz) δ ppm: 56.5 (2OCH_3_), 61.3 (OCH_3_), 105.9 (C), 108.7 (2CH), 115.6 (2CH), 116.0 (CN), 120.4 (2CH), 124.3 (2CH), 126.4 (2C), 126.9 (C), 127.3 (2CH), 138.4 (2C), 143.3 (C), 153.6 (2C), 154.8 (=CH), 163.0 (C=O); IR ν (cm^−1^): 2210, 1673, 1568, 1502, 1441, 1326, 1297, 1222, 1160, 1129, 1074, 1034, 995, 938, 867, 829, 754, 643, 615. Anal. Calcd for C_25_H_20_N_2_O_4_: C, 72.80; H, 4.89; N, 6.79. Found: C, 73.11; H, 5.03; N, 6.98%.

#### 3.2.37. (E)-2-(9H-Carbazole-9-carbonyl)-3-(5-methoxy-1H-indol-3-yl)acrylonitrile (**3o**)

General procedure A was used with 3-(9*H*-carbazol-9-yl)-3-oxopropanenitrile **5** (0.25 g, 1.07 mmol), 5-methoxy-1*H*-indole-3-carbaldehyde **31** (0.22 g, 1.26 mmol), piperidine (3 drops), and glacial acetic acid (1 drop) in 10 mL acetonitrile to obtain pure **3o** (0.29 g, 0.74 mmol, 69% yield) as a yellow solid; mp > 250 °C (EtOH); ^1^H NMR (CDCl_3_, 500 MHz) δ ppm: 3.85 (s, 3H, OC*H*_3_), 6.98 (d, *J* = 8.5 Hz, 1H, Ar*H*), 7.16 (s, 1H, Ar*H*), 7.33–7.51 (m, 5H, Ar*H*), 7.98 (d, *J* = 7.0 Hz, 2H, Ar*H*), 8.04 (d, *J* = 7.0 Hz, 2H, Ar*H*), 8.58 (s, 1H, =C*H*), 8.72 (s, 1H, Ar*H*), 9.21 (br s, 1H, N*H*); ^13^C NMR (CDCl_3_, 125 MHz) δ ppm: 56.0 (OCH_3_), 98.8 (C), 100.3 (CH), 112.0 (C), 113.3 (CH), 115.0 (CH), 115.4 (2CH), 116.0 (CN), 120.3 (2CH), 123.8 (2CH), 126.1 (2C), 127.1 (2CH), 128.6 (C), 130.6 (C), 131.5 (CH), 138.7 (2C), 147.9 (=CH), 156.8 (C), 164.1 (C=O); IR ν (cm^−1^): 3412, 2917, 2206, 2019, 1978, 1665, 1571, 1481, 1441, 1360, 1299, 1245, 1212, 1137, 1056, 939, 880, 827, 800, 768, 679, 618. Anal. Calcd for C_25_H_17_N_3_O_2_: C, 76.71; H, 4.38; N, 10.74. Found: C, 76.96; H, 4.59; N, 10.92%.

#### 3.2.38. (E)-2-(9H-Carbazole-9-carbonyl)-3-(5-methoxy-1-methyl-1H-indol-3-yl)acrylonitrile (**3p**)

General procedure A was used with 3-(9*H*-carbazol-9-yl)-3-oxopropanenitrile **5** (0.25 g, 1.07 mmol), 5-methoxy-1-methyl-1*H*-indole-3-carbaldehyde **32** (0.24 g, 1.27 mmol), piperidine (3 drops), and glacial acetic acid (1 drop) in 10 mL acetonitrile to obtain pure **3p** (0.37 g, 0.92 mmol, 85% yield) as a yellow solid; mp 227–229 °C (EtOH); ^1^H NMR (CDCl_3_, 500 MHz) δ ppm: 3.85 (s, 3H, NC*H*_3_), 3.92 (s, 3H, OC*H*_3_), 6.98–7.03 (m, 1H, Ar*H*), 7.14–7.18 (m, 1H, Ar*H*), 7.31 (t, *J* = 8.0 Hz, 1H, Ar*H*), 7.39 (t, *J* = 8.0 Hz, 2H, Ar*H*), 7.46 (t, *J* = 8.0 Hz, 2H, Ar*H*), 7.96 (d, *J* = 8.0 Hz, 2H, Ar*H*), 8.03 (dd, *J* = 8.0 Hz, 2H, Ar*H*), 8.55 (s, 1H, =C*H*), 8.59 (s, 1H, Ar*H*); ^13^C NMR (CDCl_3_, 125 MHz) δ ppm: 34.5 (CH_3_), 56.0 (OCH_3_), 97.0 (C), 100.5 (CH), 110.6 (C), 111.7 (CH), 114.5 (CH), 115.3 (2CH), 118.6 (CN), 120.2 (2CH), 123.5 (2CH), 126.0 (2C), 127.0 (2CH), 129.6 (C), 132.1 (C), 135.5 (CH), 138.8 (2C), 147.5 (=CH), 157.0 (C), 164.4 (C=O). IR ν (cm^−1^): 2198, 1658, 1621, 1568, 1519, 1476, 1442, 1360, 1296, 1233, 1195, 1133, 1070, 1044, 935, 862, 851, 743, 683, 641. Anal. Calcd for C_26_H_19_N_3_O_2_: C, 77.02; H, 4.72; N, 10.36. Found: C, 76.89; H, 4.56; N, 10.22%.

### 3.3. General Procedure for the Ultrasound-Mediated Synthesis of Chalcone Analogues (***1f***, ***1m***, ***2a***, ***2c-f***, ***2h-j***, and ***2o***) by Claisen–Schmidt Condensation—Procedure B

In a beaker, to a mixture of *N*-3-oxo-propanenitrile **4a** or **4b** (1 equiv.), aldehyde (1–1.2 equiv.) and lithium hydroxide (0.7 equiv.) in ethanol, at room temperature, an ultrasonic agitation was applied for 45 to 120 s (amplitude = 0.3). After cooling to rt for 1 to 5 h, a precipitate was formed, filtered, and purified by recrystallization from ethanol to obtain target cyanochalcone **1f**, **1m**, **2a**, **2c-f**, **2h-j**, or **2o** as a pure solid.

#### 3.3.1. (E)-3-(2,4-Dichlorophenyl)-2-(10H-phenothiazine-10-carbonyl)acrylonitrile (**1f**)

General procedure B was used with 3-oxo-3-(10*H*-phenothiazin-10-yl)propanenitrile **4a** (0.25 g, 0.94 mmol), 2,4-dichlorobenzaldehyde **17** (0.20 g, 1.13 mmol), and lithium hydroxide (0.02 g, 0.66 mmol) in 25 mL ethanol, at room temperature, and an ultrasonic agitation was applied for 120 s (amplitude = 0.3; t_i_ = 20 °C; t_f_ = 50 °C; E = 539 J). After 3 h, a precipitate was formed, filtered, and purified by recrystallization from ethanol to obtain pure compound **1f** (0.30 g, 0.71 mmol, 75% yield) as a yellow solid; mp 176–178 °C (EtOH); ^1^H NMR (CDCl_3_, 400 MHz) δ ppm: 7.27–7.39 (m, 5H, Ar*H*), 7.48–7.52 (m, 3H, Ar*H*), 7.62 (d, *J* = 8.0 Hz, 2H, Ar*H*), 7.92 (d, *J* = 8.0 Hz, 1H, Ar*H*), 8.20 (s, 1H, C*H*=); ^13^C NMR (CDCl_3_, 100 MHz) δ ppm: 110.7 (C), 113.5 (CN), 126.3 (2CH), 127.4 (2CH), 127.6 (2CH), 127.8 (CH), 128.2 (2CH), 128.9 (C), 130.1 (C), 130.1 (CH), 132.7 (2C), 136.3 (C), 137.9 (CH), 138.6 (2C), 148.1 (=CH), 161.2 (C=O); IR ν (cm^−1^): 2205, 1660, 1579, 1479, 1460, 1325, 1291, 1263, 1239, 1196, 1155, 1100, 1029, 960, 927, 865, 825, 753, 728, 653. Anal. Calcd for C_22_H_12_Cl_2_N_2_OS: C, 62.42; H, 2.86; N, 6.62. Found: C, 62.37; H, 2.94; N, 6.80%.

#### 3.3.2. (E)-2-(10H-Phenothiazine-10-carbonyl)-3-(3,4,5-trimethoxyphenyl)acrylonitrile (**1m**)

General procedure A was used with 3-oxo-3-(10*H*-phenothiazin-10-yl)propanenitrile **4a** (0.50 g, 1.87 mmol), 3,4,5-trimethoxybenzaldehyde **27** (0.44 g, 2.26 mmol), piperidine (4 drops), and glacial acetic acid (3 drops) in 15 mL ethanol to obtain pure **1m** (0.55 g, 1.24 mmol, 74% yield) as a yellow solid.

General procedure B was used with 3-oxo-3-(10*H*-phenothiazin-10-yl)propanenitrile **4a** (0.50 g, 1.87 mmol), 3,4,5-trimethoxybenzaldehyde **27** (0.44 g, 2.26 mmol), and lithium hydroxide (0.03 g, 1.25 mmol) in 30 mL ethanol, at room temperature, and an ultrasonic agitation was applied for 120 s (amplitude = 0.3; t_i_ = 19 °C; t_f_ = 52 °C; E = 575 J). After 1 h, a precipitate was formed, filtered, and purified by recrystallization from ethanol to obtain pure compound **1m** (0.61 g, 1.38 mmol, 74% yield) as a yellow solid; mp 192–194 °C (EtOH); ^1^H NMR (CDCl_3_, 400 MHz) δ ppm: 3.86 (s, 6H, 2OC*H*_3_), 3.92 (s, 3H, OC*H*_3_), 7.14 (s, 2H, Ar*H*), 7.22–7.38 (m, 4H, Ar*H*), 7.50 (d, *J* = 7.6 Hz, 2H, Ar*H*), 7.62 (d, *J* = 7.6 Hz, 2H, Ar*H*), 7.99 (s, 1H, =C*H*); ^13^C NMR (CDCl_3_, 100 MHz) δ ppm: 56.3 (2OCH_3_), 61.1 (OCH_3_), 105.5 (C), 108.0 (2CH), 114.6 (CN), 126.3 (2CH), 127.2 (C), 127.3 (2CH), 127.4 (2CH), 128.1 (2CH), 132.8 (2C), 138.3 (2C), 142.2 (C), 153.2 (2C), 154.1 (=CH), 162.1 (C=O); IR ν (cm^−1^): 2218, 1662, 1462, 1316, 1258, 1152, 812, 745. Anal. Calcd for C_25_H_20_N_2_O_4_S: C, 67.55; H, 4.54; N, 6.30. Found: C, 67.41; H, 4.36; N, 6.68%.

#### 3.3.3. (E)-2-(2-(Methylthio)-10H-phenothiazine-10-carbonyl)-3-(p-tolyl)acrylonitrile (**2a**)

General procedure B was used with 3-(2-(methylthio)-10*H*-phenothiazin-10-yl)-3-oxopropanenitrile **4b** (0.40 g, 1.28 mmol), 4-methylbenzaldehyde **9** (0.17 g, 1.41 mmol), and lithium hydroxide (0.02 g, 0.84 mmol) in 30 mL ethanol, at room temperature, and an ultrasonic agitation was applied for 45 s (amplitude = 0.3; t_i_ = 19 °C; t_f_ = 35 °C; E = 125 J). After cooling to rt for 1 h, the formed precipitate was filtered and washed with water and ethanol to obtain pure **2a** (0.35 g, 0.87 mmol, 68% yield) as a green solid; mp 129–131 °C (EtOH); ^1^H NMR (CDCl_3_, 400 MHz) δ ppm: 2.40 (s, 3H, C*H*_3_), 2.45 (s, 3H, SC*H*_3_), 7.16 (dd, *J* = 8.0, 2.0 Hz, 1H, Ar*H*), 7.22–7.34 (m, 4H, Ar*H*), 7.36 (d, *J* = 8.0 Hz, 1H, Ar*H*), 7.48 (dd, *J* = 8.0, 1.6 Hz, 1H, Ar*H*), 7.53 (dd, *J* = 8.0, 2.0 Hz, 1H, Ar*H*), 7.59 (d, *J* = 1.6 Hz, 1H, Ar*H*), 7.73 (d, *J* = 8.0 Hz, 2H, Ar*H*), 8.00 (s, 1H, =C*H*); ^13^C NMR (CDCl_3_, 100 MHz) δ ppm: 16.2 (SCH_3_), 21.8 (CH_3_), 105.7 (C), 114.4 (C≡N), 124.4 (CH), 125.7 (CH), 126.2 (CH), 127.3 (CH), 127.4 (CH), 127.9 (CH), 128.1 (CH), 128.9 (C), 129.4 (C), 129.9 (2CH), 130.5 (2CH), 132.9 (C), 138.2 (C), 138.3 (C), 138.8 (C), 143.8 (C), 154.0 (=CH), 162.2 (C=O); IR ν (cm^−1^): 2212, 1662, 1587, 1460, 1400, 1329, 1262, 1183, 1105, 1032, 949, 868, 806, 748,665. Anal. Calcd for C_24_H_18_N_2_OS_2_: C, 69.54; H, 4.38; N, 6.76. Found: C, 69.71; H, 4.57; N, 6.95%.

#### 3.3.4. (E)-4-(2-Cyano-3-(2-(methylthio)-10H-phenothiazin-10-yl)-3-oxoprop-1-en-1-yl)benzonitrile (**2c**)

General procedure B was used with 3-(2-(methylthio)-10*H*-phenothiazin-10-yl)-3-oxopropanenitrile **4b** (0.40 g, 1.28 mmol), 4-cyanobenzaldehyde **13** (0.21 g, 1.60 mmol), and lithium hydroxide (0.02 g, 0.84 mmol) in 30 mL ethanol, at room temperature, and an ultrasonic agitation was applied for 60 s (amplitude = 0.3; t_i_ = 18 °C; t_f_ = 41 °C; E = 169 J). After cooling to rt for 2 h, the formed precipitate was filtered, washed with water and ethanol, and then purified by recrystallization from ethanol to give pure **2c** (0.43 g, 0.1 mmol, 78% yield) as a yellow solid; mp 207–209 °C (EtOH); ^1^H NMR (CDCl_3_, 400 MHz) δ ppm: 2.47 (s, 3H, SC*H*_3_), 7.19 (dd, *J* = 8.4, 2.0 Hz, 1H, Ar*H*), 7.29–7.36 (m, 2H, Ar*H*), 7.40 (d, *J* = 8.4 Hz, 1H, Ar*H*), 7.48–7.57 (m, 3H, Ar*H*), 7.73 (dd, *J* = 8.0, 2.0 Hz, 2H, Ar*H*), 7.87 (d, *J* = 8.0 Hz, 2H, Ar*H*), 8.01 (s, 1H, =C*H*); ^13^C NMR (CDCl_3_, 100 MHz,) δ ppm: 16.1 (SCH_3_), 111.0 (C), 113.3 (C≡N), 115.4 (C≡N), 117.8 (C), 124.2 (CH), 125.7 (CH), 126.1 (CH), 127.4 (CH), 127.8 (CH), 128.1 (CH), 128.3 (CH), 128.7 (C), 130.3 (2CH), 132.8 (2CH), 133.0 (C), 135.8 (C), 137.7 (C), 138.2 (C), 138.6 (C), 151.0 (=CH), 160.9 (C=O); IR ν (cm^−1^): 2227, 1670, 1459, 1329, 1194, 1108, 840, 798, 746, 664. Anal. Calcd for C_24_H_15_N_3_OS_2_: C, 67.74; H, 3.55; N, 9.87. Found: C, 68.11; H, 3.68; N, 10.03%.

#### 3.3.5. (E)-2-(2-(Methylthio)-10H-phenothiazine-10-carbonyl)-3-(4-(trifluoromethyl)phenyl)acrylonitrile (**2d**)

General procedure B was used with 3-(2-(methylthio)-10*H*-phenothiazin-10-yl)-3-oxopropanenitrile **4b** (0.40 g, 1.28 mmol), 4-(trifluoromethyl)benzaldehyde **14** (0.25 g, 1.44 mmol), and lithium hydroxide (0.02 g, 0.84 mmol) in 30 mL ethanol, at room temperature, and an ultrasonic agitation was applied for 60 s (amplitude = 0.3; t_i_ = 19 °C; t_f_ = 41 °C; E = 158 J). A precipitate formed immediately after and was filtered and washed with water and ethanol to afford pure **2d** (0.13 g, 0.28 mmol, 22% yield) as a green-yellow solid. After 3h at rt, the filtrate precipitated, and the precipitate was then filtered and purified by recrystallization from ethanol to give additional mass of pure **2d** (total 0.37 g, 0.79 mmol, 61% yield) as a green-yellow solid; mp 167–169 °C (EtOH); ^1^H NMR (CDCl_3_, 400 MHz,) δ ppm: 2.47 (s, 3H, SC*H*_3_), 7.18 (dd, *J* = 8.4, 2.0 Hz, 1H, Ar*H*), 7.28–7.36 (m, 2H, Ar*H*), 7.39 (d, *J* = 8.4 Hz, 1H, Ar*H*), 7.48–7.58 (m, 3H, Ar*H*), 7.70 (d, *J* = 8.4 Hz, 2H, Ar*H*), 7.89 (d, *J* = 8.4 Hz, 2H, Ar*H*), 8.03 (s, 1H, =C*H*); ^13^C NMR (CDCl_3_, 100 MHz) δ ppm: 16.1 (SCH_3_), 110.1 (C), 113.5 (C≡N), 122.0 (C), 124.3 (CH), 124.7 (C), 125.7 (CH), 126.0 (CH), 126.1 (CH), 126.2 (CH), 127.4 (CH), 127.8 (CH), 128.1 (CH), 128.3 (CH), 128.8 (C), 130.3 (2CH), 133.4 (t, *J* = 39.5 Hz, CF_3_), 135.1 (C), 137.8 (C), 138.4 (C), 138.5 (C), 151.7 (=CH), 161.2 (C=O); IR ν (cm^−1^): 2365, 1671, 1458, 1317, 1160, 1109, 1067, 929, 806, 748, 642. Anal. Calcd for C_24_H_15_F_3_N_2_OS_2_: C, 61.53; H, 3.23; N, 5.98. Found: C, 61.27; H, 3.08; N, 5.76%.

#### 3.3.6. (E)-3-(3-Fluoro-4-methoxyphenyl)-2-(2-(methylthio)-10H-phenothiazine-10-carbonyl)acrylonitrile (**2e**)

General procedure B was used with 3-(2-(methylthio)-10*H*-phenothiazin-10-yl)-3-oxopropanenitrile **4b** (0.40 g, 1.28 mmol), 3-fluoro-4-trimethoxybenzaldehyde **21** (0.24 g, 1.53 mmol), and lithium hydroxide (0.02 g, 0.84 mmol) in 30 mL ethanol, at room temperature, and an ultrasonic agitation was applied for 60 s (amplitude = 0.3; t_i_ = 19 °C; t_f_ = 45 °C; E = 160 J). A precipitate formed immediately after and was filtered and washed with water and ethanol to afford the pure compound **2e** (0.42 g, 0.90 mmol, 74% yield) as a yellow solid; mp 178–180 °C (EtOH); ^1^H NMR (CDCl_3_, 400 MHz) δ ppm: 2.46 (s, 3H, SC*H*_3_), 3.95 (s, 3H, OC*H*_3_), 7.00 (t, *J* = 8.0 Hz, 1H, Ar*H*), 7.17 (dd, *J* = 8.0, 2.0 Hz, 1H, Ar*H*), 7.26–7.35 (m, 2H, Ar*H*), 7.37 (d, *J* = 8.0 Hz, 1H, Ar*H*), 7.47–7.54 (m, 2H, Ar*H*), 7.57–7.63 (m, 2H, Ar*H*), 7.66 (dd, *J* = 12.0, 2.0 Hz, 1H, Ar*H*), 7.94 (s, 1H, =C*H*); ^13^C NMR (CDCl_3_, 100 MHz) δ ppm: 16.2 (SCH_3_), 56.3 (OCH_3_), 105.2 (C), 113.1 (d, *J* = 1.5 Hz, CH), 114.2 (CN), 117.4 (d, *J* = 19.7 Hz, CH), 124.4 (CH), 125.1 (d, *J* = 7.6 Hz, C), 125.6 (CH), 126.2 (CH), 127.3 (CH), 127.5 (CH), 128.0 (CH), 128.2 (CH), 128.5 (d, *J* = 3.0 Hz, CH), 128.9 (C), 133.0 (C), 138.1 (C), 138.3 (C), 138.7 (C), 151.5 (d, *J* = 10.7 Hz, C), 151.9 (d, *J* = 247.5 Hz, CF_3_), 152.4 (d, *J* = 2.2 Hz, =CH), 162.0 (C=O); IR ν (cm^−1^): 2213, 1674, 1598, 1514, 1320, 1288, 1256, 1140, 1018, 816, 752. Anal. Calcd for C_24_H_17_FN_2_O_2_S_2_: C, 64.27; H, 3.82; N, 6.25. Found: C, 64.38; H, 3.90; N, 6.44%.

#### 3.3.7. (E)-3-(Benzo[d][1,3]dioxol-5-yl)-2-(2-(methylthio)-10H-phenothiazine-10-carbonyl)acrylonitrile (**2f**)

General procedure B was used with 3-(2-(methylthio)-10*H*-phenothiazin-10-yl)-3-oxopropanenitrile **4b** (0.40 g, 1.28 mmol), 1,3-benzodioxole-5-carboxaldehyde **28** (0.21 g, 1.41 mmol), and lithium hydroxide (0.02 g, 0.84 mmol) in 30 mL ethanol, at room temperature, and an ultrasonic agitation was applied for 60 s (amplitude = 0.3; t_i_ = 18 °C; t_f_ = 45 °C; E = 149 J). A precipitate formed immediately after and was filtered and washed with water and ethanol to afford the pure compound **2f** (0.38 g, 0.85 mmol, 67% yield) as a yellow-green solid; mp 192–194 °C (EtOH); ^1^H NMR (CDCl_3_, 400 MHz) δ ppm: 2.46 (s, 3H, SC*H*_3_), 6.05 (s, 2H, C*H*_2_), 6.86 (d, *J* = 8.4 Hz, 1H, Ar*H*), 7.16 (dd, *J* = 8.4, 2.0 Hz, 1H, Ar*H*), 7.26–7.38 (m, 4H, Ar*H*), 7.47–7.55 (m, 3H, Ar*H*), 7.59 (d, *J* = 2.0 Hz, 1H, Ar*H*), 7.93 (s, 1H, =C*H*); ^13^C NMR (CDCl_3_, 100 MHz) δ ppm: 16.2 (SCH_3_), 102.1 (CH_2_), 104.0 (C), 108.6 (CH), 108.8 (CH), 114.5 (C≡N), 124.4 (CH), 125.6 (CH), 126.2 (CH), 126.4 (C), 127.3 (CH), 127.4 (CH), 127.9 (CH), 128.1 (CH), 128.6 (CH), 128.9 (C), 133.0 (C), 138.3 (2C), 138.8 (C), 148.5 (C), 151.7 (C), 153.6 (=CH), 162.3 (C=O); IR ν (cm^−1^): 2214, 1671, 1584, 1445, 1310, 1256, 1036, 920, 810, 753, 625. Anal. Calcd for C_24_H_16_N_2_O_3_S_2_: C, 64.85; H, 3.63; N, 6.30. Found: C, 64.72; H, 3.55; N, 6.12%.

#### 3.3.8. (E)-3-(2,4-Dichlorophenyl)-2-(2-(methylthio)-10H-phenothiazine-10-carbonyl)acrylonitrile (**2h**)

General procedure B was used with 3-(2-(methylthio)-10*H*-phenothiazin-10-yl)-3-oxopropanenitrile **4b** (0.34 g, 1.09 mmol), 2,4-dichlorobenzaldehyde **17** (0.19 g, 1.09 mmol), and lithium hydroxide (0.02 g, 0.84 mmol) in 25 mL ethanol, at room temperature, and an ultrasonic agitation was applied for 90 s (amplitude = 0.3; t_i_ = 20 °C; t_f_ = 59 °C; E = 282 J). After 3 h, a precipitate formed and was filtered and purified by recrystallization from ethanol to obtain the pure compound **2h** (0.34 g, 0.72 mmol, 67% yield) as a yellow solid; mp 176–178 °C (EtOH); ^1^H NMR (CDCl_3_, 500 MHz) δ ppm: 2.47 (s, 3H, SC*H*_3_), 7.16 (dd, *J* = 8.0, 2.0 Hz, 1H, Ar*H*), 7.27–7.36 (m, 3H, Ar*H*), 7.38 (d, *J* = 7.5 Hz, 1H, Ar*H*), 7.45–7.51 (m, 2H, Ar*H*), 7.55 (s, 2H, Ar*H*), 7.90 (d, *J* = 8.5 Hz, 1H, Ar*H*), 8.20 (s, 1H, C*H*=); ^13^C NMR (CDCl_3_, 125 MHz) δ ppm: 16.2 (SCH_3_), 110.9 (C), 113.6 (C≡N), 124.2 (CH), 125.8 (CH), 126.3 (CH), 127.5 (CH), 127.8 (CH), 128.0 (CH), 128.2 (CH), 128.3 (CH), 128.8 (C), 129.0 (C), 130.2 (2CH), 133.0 (C), 136.3 (C), 137.9 (C), 138.5 (C), 138.7 (2C), 148.3 (=CH), 161.3 (C=O); IR ν (cm^−1^): 2214, 1670, 1456, 1396, 1327, 1262, 1241, 1192, 1103, 1047, 972, 828, 793, 744, 728, 695. Anal. Calcd for C_23_H_14_Cl_2_N_2_OS_2_: C, 58.85; H, 3.01; N, 5.97. Found: C, 59.14; H, 3.26; N, 6.25%.

#### 3.3.9. (E)-2-(2-(Methylthio)-10H-phenothiazine-10-carbonyl)-3-(3,4,5-trimethoxyphenyl)acrylonitrile (**2i**)

General procedure B was used with 3-(2-(methylthio)-10*H*-phenothiazin-10-yl)-3-oxopropanenitrile **4b** (0.25 g, 0.80 mmol), 3,4,5-trimethoxybenzaldehyde **27** (0.16 g, 0.82 mmol), and lithium hydroxide (0.02 g, 0.84 mmol) in 25 mL ethanol, at room temperature, and an ultrasonic agitation was applied for 90 s (amplitude = 0.3; t_i_ = 20 °C; t_f_ = 59 °C; E = 343 J). After 3 h, the formed precipitate was filtered and purified by recrystallization from ethanol to obtain the pure compound **2i** (0.28 g, 0.57 mmol, 72% yield) as a yellow solid; mp 198–199 °C (EtOH); ^1^H NMR (CDCl_3_, 500 MHz) δ ppm: 2.44 (s, 3H, SC*H*_3_), 3.76 (s, 3H, OC*H*_3_), 3.78 (s, 6H, 2OC*H*_3_), 7.25 (s, 2H, Ar*H*), 7.31–7.78 (m, 7H, Ar*H*), 8.12 (s, 1H, =C*H*); ^13^C NMR (CDCl_3_, 125 MHz) δ ppm: 15.0 (SCH_3_), 56.0 (2OCH_3_), 60.3 (OCH_3_), 105.1 (C), 107.8 (2CH), 114.5 (C≡N), 123.9 (CH), 125.2 (CH), 126.6 (CH), 127.2 (C), 127.5 (CH), 127.7 (CH), 127.8 (C), 128.0 (CH), 128.1 (CH), 131.8 (C), 137.6 (C), 138.1 (C), 138.3 (C), 141.3 (C), 152.9 (2C), 154.3 (=CH), 161.2 (C=O); IR ν (cm^−1^): 2208, 1672, 1573, 1504, 1461, 1421, 1394, 1308, 1292, 1258, 1241, 1186, 1161, 1133, 1110, 991, 937, 928, 863, 808, 754, 635. Anal. Calcd for C_26_H_22_N_2_O_4_S_2_: C, 63.65; H, 4.52; N, 5.71. Found: C, 63.79; H, 4.68; N, 5.94%.

#### 3.3.10. (E)-2-(2-(Methylthio)-10H-phenothiazine-10-carbonyl)-3-(thiophen-2-yl)acrylonitrile (**2j**)

General procedure B was used with 3-(2-(methylthio)-10*H*-phenothiazin-10-yl)-3-oxopropanenitrile **4b** (0.40 g, 1.28 mmol), thiophene-2-carboxaldehyde **29** (0.16 g, 1.43 mmol), and lithium hydroxide (0.02 g, 0.84 mmol) in 30 mL ethanol, at room temperature, and an ultrasonic agitation was applied for 60 s (amplitude = 0.3; t_i_ = 19 °C; t_f_ = 44 °C; E = 169 J). The formed precipitate 3 h after reaction was filtered, washed with water, and purified by recrystallization from ethanol to give the pure compound **2j** (0.40 g, 0.98 mmol, 77% yield) as a yellow solid; mp 160–162 °C (EtOH); ^1^H NMR (CDCl_3_, 400 MHz) δ ppm: 2.47 (s, 3H, SC*H*_3_), 7.16–7.20 (m, 2H, Ar*H*), 7.27–7.34 (m, 2H, Ar*H*), 7.37 (d, *J* = 8.4 Hz, 1H, Ar*H*), 7.47–7.55 (m, 2H, Ar*H*), 7.60 (d, *J* = 2.0 Hz, 1H, Ar*H*), 7.68–7.71 (m, 1H, Ar*H*), 7.74 (dd, *J* = 3.6, 2.0 Hz, 1H, Ar*H*), 8.28 (s, 1H, =C*H*); ^13^C NMR (CDCl_3_, 100 MHz) δ ppm: 16.3 (SCH_3_), 103.0 (C), 114.3 (C≡N), 124.5 (CH), 125.8 (CH), 126.2 (CH), 127.3 (CH), 127.5 (CH), 127.9 (CH), 128.2 (CH), 128.4 (CH), 128.9 (C), 133.0 (C), 134.1 (CH), 136.1 (CH), 136.4 (C), 138.2 (C), 138.3 (C), 138.8 (C), 146.7 (=CH), 161.7 (C=O); IR ν (cm^−1^): 2216, 1672, 1593, 1458, 1301, 1192, 1109, 943, 817, 755, 719. Anal. Calcd for C_21_H_14_N_2_OS_3_: C, 62.04; H, 3.47; N, 6.89. Found: C, 62.35; H, 3.58; N, 7.06%.

#### 3.3.11. (E)-2-(2-(Methylthio)-10H-phenothiazine-10-carbonyl)-5-phenylpenta-2,4-dienenitrile (**2o**)

General procedure B was used with 3-(2-(methylthio)-10*H*-phenothiazin-10-yl)-3-oxopropanenitrile **4b** (0.40 g, 1.28 mmol), cinnamaldehyde **36** (0.19 g, 1.41 mmol), and lithium hydroxide (0.02 g, 0.84 mmol) in 30 mL ethanol, at room temperature, and an ultrasonic agitation was applied for 60 s (amplitude = 0.3; t_i_ = 19 °C; t_f_ = 44 °C; E = 166 J). The precipitate formed immediately after was filtered and washed with water and ethanol to afford the pure compound **2o** (0.22 g, 0.52 mmol, 40% yield) as an orange solid. After 3h at rt, the filtrate precipitated, and the precipitate was then filtered and purified by recrystallization from ethanol to give additional mass of pure **2o** (total 0.33 g, 0.77 mmol, 61% yield) as an orange solid; mp 161–163 °C (EtOH); ^1^H NMR (400 MHz, CDCl_3_) δ ppm: 2.48 (d, *J* = 1.2 Hz, 3H, SC*H*_3_), 7.15–7.20 (m, 3H, =C*H*+2Ar*H*), 7.26–7.42 (m, 6H, Ar*H*), 7.47–7.57 (m, 4H, =C*H*+3Ar*H*), 7.58 (d, *J* = 2.0 Hz, 1H, Ar*H*), 7.94 (dd, *J* = 6.4, 4.8 Hz, 1H, =C*H*); ^13^C NMR (100 MHz, CDCl_3_) δ ppm: 16.2 (SCH_3_), 107.9 (C), 113.1 (CN), 123.3 (CH), 124.4 (CH), 125.6 (CH), 126.3 (CH), 127.3 (CH), 127.5 (CH), 127.9 (CH), 128.1 (CH), 128.4 (2CH), 128.8 (C), 129.1 (2CH), 130.9 (CH), 133.0 (C), 134.8 (C), 138.1 (C), 138.3 (C), 138.7 (C), 147.4 (=CH), 155.3 (=CH), 161.3 (C=O); IR ν (cm^−1^): 2212, 1674, 1562, 1443, 1298, 1167, 980, 819, 745, 687. Anal. Calcd for C_25_H_18_N_2_OS_2_: C, 70.39; H, 4.25; N, 6.57. Found: C, 70.48; H, 4.47; N, 6.81%.

### 3.4. Human FTase Assay

Assays were realized in 96-well plates, prepared with a Biomek NKMC and a Biomek 3000 from Beckman Coulter, and read on a Wallac Victor fluorimeter from PerkineElmer [28]. Per well, 20 μL of farnesyl pyrophosphate (10 μM) was added to 180 μL of a solution containing 2 μL of varied concentrations of potential inhibitors (dissolved in DMSO) and 178 μL of a solution composed of 10 μL of partially purified recombinant human FTase (5 mg/mL) and 1.0 mL of Dansyl-GCVLS peptide (in the following buffer: 5.6 mM DTT, 5.6 mM MgCl_2_, 12 μM ZnCl_2_ and 0.2% (*w*/*v*) octyl-s-D-glucopyranoside, 52 mM Tris/HCl, pH 7.5). Fluorescence was recorded for 15 min (0.7 s per well, 20 repeats) at 30 °C with an excitation filter of 340 nm and an emission filter of 486 nm. Each measurement was reproduced twice (two independent experiments on different 96-well plates) in duplicate. The kinetic experiments were realized under the same conditions, either with FPP as varied substrate with a constant concentration of Dns-GCVLS of 2.5 μM or with Dns-GCVLS as varied substrate with a constant concentration of FPP of 10 μM. Nonlinear regressions were performed with Excel software.

### 3.5. Tubulin Polymerization Assay

Sheep brain tubulin was purified according to the method of Shelanski [29] by two cycles of assembly–disassembly and then dissolved in the assembly buffer containing 0.1 M MES, 0.5 mM MgCl_2_, 1 mM EGTA, and 1 mM of GTP (pH 6.6) to give a tubulin concentration of about 2–3 mg/mL. Tubulin assembly was monitored by fluorescence according to reported procedure [30] using DAPI as fluorescent molecule. Assays were realized on 96-well plates prepared with Biomek NKMC and Biomek 3000 from Beckman coulter (Villepinte, France) and read at 37 °C on Wallac Victor fluorimeter from Perkin–Elmer (Villebon-sur-Yvette, France). The IC_50_ value of each compound was determined as tubulin polymerization inhibition by 50% compared to the rate in the absence of compound. The IC_50_ values for all compounds were compared to the IC_50_ values of phenstatin and (-)-desoxypodophyllotoxin and were measured the same day under the same conditions.

### 3.6. Cell Proliferation Assay

Compounds **2k**, **2l**, and **2o** were tested on a panel of 60 human cancer cell lines at the National Cancer Institute, Germantown, MD [31]. The cytotoxicity studies were conducted using a 48h exposure protocol using the sulforhodamine B assay [32,33].

## Data Availability

The data presented in this study are available on request from the corresponding authors. Supplementary materials include copies of NMR spectra of all synthesized compounds.

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
