# Peer review of "Phenothiazine- and Carbazole-Cyanochalcones as Dual Inhibitors of Tubulin Polymerization and Human Farnesyltransferase"

_pharmaceuticals, 2023, doi:10.3390/ph16060888_

Round 1

Reviewer 1 Report

The manuscript (ID pharmaceuticals-2433650) by Andreea ZubaÈ™ et al., "Phenothiazine- and carbazole-cyanochalcones as dual inhibitors of tubulin polymerization and human farnesyltransferase", describes synthesis and structure elucidation of forty-nine cyanochalcones, which were screened in vitro for their inhibitory activity against human farnesyltransferase and tubulin polymerization, while cancer cell growth was evaluated only for 3 selected compounds.

Chemical and biological experiments are well designed and performed, the paper is nicely written, synthetically very interesting, and the characterization of the compounds is well described. English spelling and grammar are at a satisfactory level. Therefore, I could recommend publishing paper in the Pharmaceuticals, after the authors address a few minor points.

1. It is not necessary for Fig. 4 to be in the manuscript, as all data are already presented in Table 3. Figure 4 could be placed in the Supplemental data. The legend in Fig. 4 is not fully visible, some words are covered. 2. I suggest that the synthetic procedures be transferred to the Supplemental data, so that they do not burden the main manuscript body.

3. The self-citation rate is too high (about 30%), although it should not exceed 15%. That should be corrected.

Author Response

Response to reviewers

We are sincerely grateful to the reviewers who constructively evaluated our research article.

Reviewer #1:

The manuscript (ID pharmaceuticals-2433650) by Andreea ZubaÈ™ et al., "Phenothiazine- and carbazole-cyanochalcones as dual inhibitors of tubulin polymerization and human farnesyltransferase", describes synthesis and structure elucidation of forty-nine cyanochalcones, which were screened in vitro for their inhibitory activity against human farnesyltransferase and tubulin polymerization, while cancer cell growth was evaluated only for 3 selected compounds.

Chemical and biological experiments are well designed and performed, the paper is nicely written, synthetically very interesting, and the characterization of the compounds is well described. English spelling and grammar are at a satisfactory level. Therefore, I could recommend publishing paper in the Pharmaceuticals, after the authors address a few minor points.

  1. It is not necessary for Fig. 4 to be in the manuscript, as all data are already presented in Table 3. Figure 4 could be placed in the Supplemental data. The legend in Fig. 4 is not fully visible, some words are covered.

The previous Figure 4 has been moved to the Supplementary data section as Figure S1.

  1. I suggest that the synthetic procedures be transferred to the Supplemental data, so that they do not burden the main manuscript body.

We have decided to keep the synthetic procedures in the main text of the manuscript which are useful for synthetic chemists. In the supplementary data, we have provided all the copies of the NMR and IR spectra.

  1. The self-citation rate is too high (about 30%), although it should not exceed 15%. That should be corrected.

This was corrected. We have only kept 6 references from our team in the corrected manuscript over 33 references.

Reviewer 2 Report

The manuscript pharmaceuticals-2433650 "Phenothiazine- and carbazole-cyanochalcones as dual inhibitors of tubulin polymerization and human farnesyltransferase" by ZubaÈ™ et al. is describes the synthesis of a large library of phenothiazine- and carbazole-cyanochalcones derivatives and the study of their antitumor potential. The synthesis was confirmed by 1H, 13C, IR spectroscopy and elemental analysis. The authors obtained interesting SARs, so I think this paper will be of interest to the readers of Pharmaceuticals.

However, I have some questions and comments.

1) I recommend the authors to pay attention to the correlations between the obtained compounds structure and their spectral characteristics (NMR spectra, etc.). This will be useful for synthetic chemists.

2) Please add the images of IR spectra in supplementary materials. Why did the authors not record the mass spectra of novel compounds?

3) The information in Figure 4 and Table 3 is duplicated. Something needs to be moved to supplementary materials.

4) I recommend the authors to strengthen the Introduction part about synthesis and applications of phenothiazine derivatives. Recent review articles of 2022-2023 on this topic should be added. For example, 10.3390/molecules27010276, 10.1016/j.dyepig.2022.110806, 10.1039/D2TC02085H.

5) I recommend that the authors do molecular docking to confirm the mechanism of inhibition.

6) The antitumor activity data are useless when the toxicity to healthy cells is high. What about the toxicity of the obtained compounds?

7) Please add a comparison of the results obtained with the literature, if any.

8) Minor changes:

- The text in the Figure 1 is too small.

- Some of the text is not visible in the Figure 4.

- Please indicate the E-isomer in the names of the compounds.

Please pay attention to the correct use of articles (sometimes it is missing). Please re-check English.

Author Response

We are sincerely grateful to the reviewers who constructively evaluated our research article.

Reviewer #2:

The manuscript pharmaceuticals-2433650 "Phenothiazine- and carbazole-cyanochalcones as dual inhibitors of tubulin polymerization and human farnesyltransferase" by ZubaÈ™ et al. is describes the synthesis of a large library of phenothiazine- and carbazole-cyanochalcones derivatives and the study of their antitumor potential. The synthesis was confirmed by 1H, 13C, IR spectroscopy and elemental analysis. The authors obtained interesting SARs, so I think this paper will be of interest to the readers of Pharmaceuticals.

However, I have some questions and comments.

1) I recommend the authors to pay attention to the correlations between the obtained compounds structure and their spectral characteristics (NMR spectra, etc.). This will be useful for synthetic chemists.

Copies of two-dimensional nuclear magnetic resonance spectroscopy (2D NMR) correlations for compounds 1g, 2l and 3k have been provided in the revised Supplementary material section which are very useful for synthetic chemists. However, the structure of cyano-chalcones is quite simple 1 and the spectra are easy to interpret.

2) Please add the images of IR spectra in supplementary materials. Why did the authors not record the mass spectra of novel compounds?

The images of the IR spectra have been added in the Supplementary material section.

We did not have access to a HRMS equipment to record the mass spectra of compounds.

3) The information in Figure 4 and Table 3 is duplicated. Something needs to be moved to supplementary materials.

The reviewer is right. The Figure 4 has been moved to the Supplementary data section and Table 3 has been conserved in the current revised manuscript.

4) I recommend the authors to strengthen the Introduction part about synthesis and applications of phenothiazine derivatives. Recent review articles of 2022-2023 on this topic should be added. For example, 10.3390/molecules27010276, 10.1016/j.dyepig.2022.110806, 10.1039/D2TC02085H.

The suggested references have been added in the Introduction of the revised manuscript as references 21-23.

5) I recommend that the authors do molecular docking to confirm the mechanism of inhibition.

The molecular docking was performed on the eight dual FTIs/MTIs inhibitors on tubulin and FTase binding sites. The information is available in the revised version of the manuscript and in the Supplementary data section.

6) The antitumor activity data are useless when the toxicity to healthy cells is high. What about the toxicity of the obtained compounds?

At this early-stage development, we did not perform the toxicity of the cyano-chalcones on “healthy cells”. We will perform this evaluation at the same time as the evaluation of the pharmacokinetic parameters in due course.

7) Please add a comparison of the results obtained with the literature, if any.

There are no similar results available in the literature on cyano-chalcones to allow comparison of the results obtained on both biological targets of interest (FTase and tubulin). The comparison has been made with the reference molecules previously reported, especially phenstatin. The most effective molecule 3a displayed better antitubulin activity than known inhibitors previously reported phenstatin and (-)-desoxypodophyllotoxin.

8) Minor changes:

- The text in the Figure 1 is too small.

The size of the text in Figure 1 has been increased, as requested.

- Some of the text is not visible in the Figure 4.

The Figure 4 has been transferred to the supplementary data section as Figure S1 and the text is visible.

- Please indicate the E-isomer in the names of the compounds.

This has been corrected for the names of all the compounds 1-3.

Round 2

Reviewer 2 Report

I thank the authors for answering my questions and improving the manuscript